# Image Tokens Matter: Mitigating Hallucination in Discrete Tokenizer-based Large Vision-Language Models via Latent Editing

**Weixing Wang** [1][*], **Zifeng Ding** [2][*], **Jindong Gu** [3][†], **Rui Cao**[2]
**Gerard de Melo**[1], **Christoph Meinel**[4], **Haojin Yang**[1]
[1]Hasso Plattner Institute, University of Potsdam, [2]University of Cambridge
[3]University of Oxford, [4]German University of Digital Science
`weixing.wang@hpi.de,zd320@cam.ac.uk,jindong.gu@outlook.com`

## Abstract

Large Vision-Language Models (LVLMs) with discrete image tokenizers unify multimodal representations by encoding visual inputs into a finite set of tokens. Despite their effectiveness, we find that these models still hallucinate non-existent objects. We hypothesize that this may be due to visual priors induced during training: When certain image tokens frequently co-occur in the same spatial regions and represent shared objects, they become strongly associated with the verbalizations of those objects. As a result, the model may hallucinate by evoking visually absent tokens that often co-occur with present ones. To test this assumption, we construct a co-occurrence graph of image tokens using a segmentation dataset and employ a Graph Neural Network (GNN) with contrastive learning followed by a clustering method to group tokens that frequently co-occur in similar visual contexts. We find that hallucinations predominantly correspond to clusters whose tokens dominate the input, and more specifically, that the visually absent tokens in those clusters show much higher correlation with hallucinated objects compared to tokens present in the image. Based on this observation, we propose a hallucination mitigation method that suppresses the influence of visually absent tokens by modifying latent image embeddings during generation. Experiments show our method reduces hallucinations while preserving expressivity. Code is available at https://github.com/weixingW/CGC-VTD/tree/main

## 1 Introduction

Large Vision-Language Models (LVLMs) have achieved remarkable progress in understanding and generating multimodal content. However, they are still prone to hallucination, often generating descriptions of non-existent objects. To address this issue, recent efforts have explored decoding-time interventions—such as Contrastive Decoding [1, 2, 3, 4]—as well as techniques that leverage latent feature representations to detect and mitigate hallucinations [5, 6].

A promising direction in advancing LVLMs is the integration of discrete image tokenizers [7, 8, 9, 10, 11, 12, 13], which quantize continuous image features into sequences of discrete tokens using a finite codebook. These tokens are structurally analogous to text tokens, enabling unified token-based processing across modalities within a single transformer architecture [14]. Despite the potential of this approach, we find that LVLMs built with discrete image tokenizers remain susceptible to hallucinations, i.e., generating false or misleading content based on the input image and prompt, such

39th Conference on Neural Information Processing Systems (NeurIPS 2025).

---

[*]Equal contribution
[†]Corresponding author

as referencing nonexistent objects or producing incoherent responses unrelated to the visual content. Furthermore, our experiments reveal that directly applying existing hallucination mitigation methods to these models may not always be effective, underscoring the need for a more tailored investigation within this emerging paradigm.

We address this problem by analyzing the role of visual priors introduced by image token co-occurrence. Prior work [15, 16] has shown that object co-occurrence can create language priors in LVLMs, where models hallucinate objects not present in the current visual or textual input but frequently co-occurring with seen objects in the training text. We extend this insight to the visual modality by investigating how discrete image tokens, much like text tokens, may encode similar co-occurrence patterns. Since image tokenizers are trained on large-scale image datasets, their learned embeddings inevitably capture co-occurrence relationships among visual patterns, implicitly introducing visual priors. We hypothesize that these learned associations among image tokens can drive hallucinations in LVLMs.

To test our hypothesis, we first introduce **Context-Guided Clustering (CGC)** to capture the visual priors encoded in discrete image tokens. CGC constructs a co-occurrence graph using a corpus of segmented images, where each node represents a unique token from the image token vocabulary. The edges in this graph encode contextually-aware co-occurrence patterns, with edge weights determined by two factors: the spatial proximity of image tokens within the quantized image representation and their semantic association, based on whether they are used to represent the same segmented object. A Graph Neural Network (GNN) is trained on this graph using a contrastive learning objective, producing graph-based token embeddings that reflect these semantic and spatial co-occurrence patterns. These embeddings are then used to cluster image tokens using K-means [17] to form groups of tokens that frequently co-occur in localized visual contexts or correspond to related visual concepts. This process yields a structured representation of the visual priors embedded in the image token vocabulary. Using the learned clusters, we analyze how hallucinations in LVLMs are related to token co-occurrence patterns. Our key finding is that hallucinated objects are often highly related to image tokens that are absent from the current visual input but belong to the top few dominant clusters (those most frequently represented in the image). These absent tokens co-occur frequently with the present ones, and their strong association may lead the LVLM to mistakenly reference them, resulting in hallucinations.

Motivated by this observation, we propose **Visual Token Decontamination (VTD)**, a latent-space intervention that reduces the influence of the visually absent tokens identified above during autoregressive decoding. VTD projects these tokens into the model's latent space and subtracts their contributions from intermediate hidden representations, thereby mitigating their hallucinative effect.

The contribution of our work is summarized as follows.

- We identify and quantify a novel source of hallucination in LVLMs with discrete image tokenizers, showing that co-occurrence patterns among image tokens introduce visual priors that can mislead the model. To our knowledge, we are the first to uncover this phenomenon.

- We propose CGC and VTD as a two-step framework that captures visual priors and mitigates hallucinations by identifying co-occurrence-driven token patterns and suppressing their influence through targeted latent space editing.

- Extensive experiments demonstrate that our method can effectively reduce LVLM hallucinations while maintaining strong efficiency.

## 2 Related Work

**LVLMs with Discrete Image Tokenizer.** Recent advancements in LVLMs have increasingly adopted discrete image tokenization as a strategy to unify multiple modalities within a shared representation space. Chameleon [7] was among the first to integrate this paradigm, enabling the model to interpret and generate images and text in arbitrary sequences. This paradigm has since evolved, with Emu3 extending the approach by enlarging the image vocabulary and incorporating video as an additional modality [8]. Janus and Janus-Pro further introduced hybrid architectures that combine continuous and discrete image encoders for greater representational flexibility [9, 13]. Show-O also advances this line of work by integrating discrete image tokens into a unified framework that supports autoregressive text generation alongside iterative masked token prediction for images [11].

While these works demonstrate strong multimodal capabilities, they also reveal a new challenge: LVLMs with discrete image tokenizers remain susceptible to hallucinations. How to effectively mitigate hallucinations in this new class of models remains an underexplored question.

**Hallucination Mitigation Methods for LVLMs.** As LVLMs continue to advance, significant research has been devoted to understanding and mitigating their tendency to hallucinate, particularly in the form of object hallucination—generating descriptions of objects that are not present in the visual input [18, 19, 1, 15]. A major line of work attributes hallucination to biases in the language modeling component. For example, OPERA [5] identifies problematic decoding patterns and applies penalty and retrospection strategies, while LURE [15] explicitly models object co-occurrence, uncertainty, and positional factors through revision-based algorithms. Though effective, these methods rely on iterative identify-and-revise procedures and require multiple decoding passes, leading to high computational overhead. Another group of methods focuses on hallucination arising from the association between visual and textual modalities. Contrastive decoding (CD) techniques [20, 3, 4] mitigate hallucinations by perturbing the visual input and contrasting the resulting token logits. VCD [1] amplifies hallucinations via input noise and contrasts them against normal outputs, while SID [2] improves efficiency and effectiveness by adaptively focusing on part of visual input. PROJECTAWAY [6] offers a post-hoc alternative by associating hallucinations with visual embeddings and subtracting possible hallucinative text tokens from intermediate LVLM layer output. These approaches have shown strong results on earlier models like LLaVA [21], but their effectiveness on newer LVLMs with discrete image tokenizers remains unproven. In our experiments, CD-based methods often perform worse in models with unified visual and text token spaces, likely due to incompatibility with discrete token representations. Compared to these directions, visual prior-driven hallucination mitigation remains largely underexplored. OHD [22] investigates hallucinations caused by CLIP encoder biases, but requires full model retraining with dense supervision, which is computationally expensive. In contrast, our approach focuses specifically on the visual priors embedded in discrete image tokenizers. We introduce a lightweight GNN-based framework to capture image token co-occurrence patterns and identify hallucination-prone tokens before generation, enabling efficient and effective mitigation without retraining the LVLM.

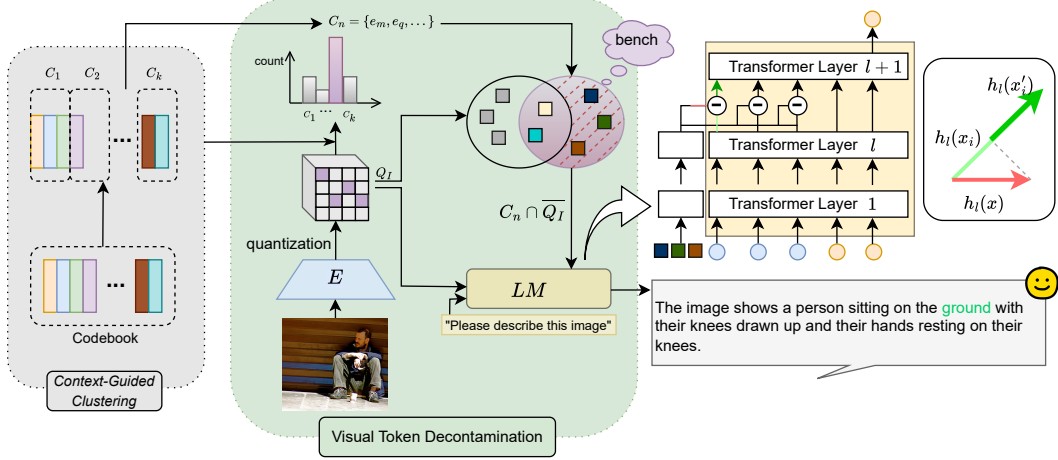

Figure 1: Overview of our proposed method, which consists of two key components: **Context-Guided Clustering (CGC)** and **Visual Token Decontamination (VTD)**. CGC organizes image tokens into clusters according to their co-occurrence patterns. VTD identifies potentially hallucinative tokens based on visual context and suppresses their influence during autoregressive generation. During inference, only VTD is applied while CGC is performed only once for every model offline.

## 3 Method

We propose a two-step approach to mitigate hallucinations in LVLMs with discrete image tokenizers. First, **Context-Guided Clustering (CGC)** captures visual priors by constructing a co-occurrence

graph of image tokens and clustering them via graph-based embeddings (Sec. 3.1). We show that hallucinations often correlate closely with the visually absent image tokens in the first few dominant clusters. (Sec. 3.2). To address this, **Visual Token Decontamination (VTD)** reduces their influence by modifying latent representations during autoregressive decoding (Sec. 3.3). See Figure 1 for an overview.

**Paradigm of LVLMs with Discrete Image Tokenizer.** We begin by defining the paradigm of LVLMs with discrete image tokenizers as follows. An input image is first encoded into a sequence of discrete image tokens $v = \{v_i\} \subseteq V$ using an image tokenizer, typically based on a Vector Quantized-Variational AutoEncoder (VQ-VAE) [23] or VQGAN [24], where the embedding of each image token corresponds to an entry in a learned codebook of size $|V|$. Similarly, the text input is tokenized into a sequence of text tokens $t = \{t_i\} \subseteq T$, where $|T|$ represents the size of the text vocabulary. For visual understanding task, the final input to the model is usually given by the concatenated image and text token sequences $x = \{v, t\}$. During inference, this token sequence is processed by a stack of $L$ transformer layers [25]. Given $x_i$, the $i$-th input token, we denote its output representation from the $l$-th transformer layer as $g^{(l)}(x_i) \in \mathbb{R}^d$, where $d$ is LVLM's hidden dimensionality. The probability distribution over the next token $y$ is computed as $P(y|x) = \text{softmax}(g^L(x_{-1})^\top \mathbf{W}\text{out})$, where $x_{-1}$ is the last token in the input and $\mathbf{W}\text{out} \in \mathbb{R}^{d \times (|V|+|T|)}$ is the output projection matrix.

## 3.1 Model Visual Priors via Context-Guided Clustering

**Capturing Visual Priors via Clustering.** In an LVLM with a discrete image tokenizer, each image token is mapped to an entry ID from a learned codebook of finite size. To maintain high compression efficiency [24], the codebook is typically limited in size (e.g., 8,192 entries in Chameleon [7]). As a result, each image token is reused to represent a wide range of visual concepts. Moreover, certain groups of image tokens frequently appear together across different images or image regions to represent common objects or scenes, depending on the visual content. These recurring patterns of co-occurrence among tokens form what we refer to as visual priors, which quantify underlining distribution bias from the image data that is used for training.

To capture visual priors, we cluster image tokens based on their co-occurrence patterns. We perform clustering based on codebook embeddings rather than directly on image token embeddings, motivated by three key observations. First, as discussed earlier, each image token in an LVLM is directly mapped from a unique codebook entry ID. This means that clustering through codebook entries (each entry in a codebook is an embedding) is equivalent to clustering image tokens. For simplicity, we henceforth refer to the co-occurrence and clustering of image tokens, with the understanding that these operations are potentially performed via their corresponding codebook entries. Second, visual priors emerge when certain groups of image tokens frequently co-occur across different images or visual regions to represent common objects or scenes. Clustering codebook entries that tend to co-occur in such contexts allows us to expose these priors. This enables us to use the image tokens present in a given input to identify other tokens—absent from the image but belonging to the same clusters—that frequently co-occur with the observed ones and may influence the model's behavior due to learned visual priors.

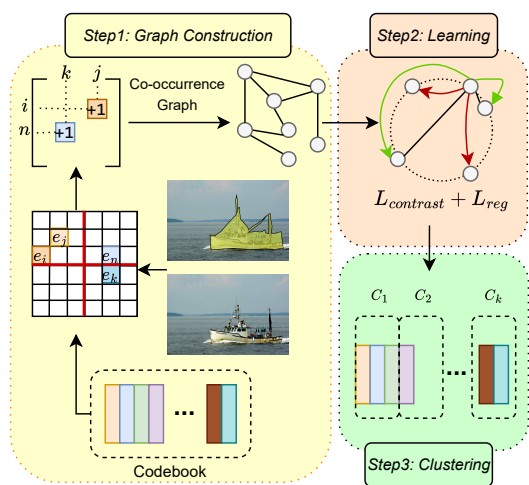

Figure 2: Illustration of the CGC pipeline. Codebook entries are first used to help construct a co-occurrence graph over image tokens, followed by a learning process to obtain graph-based embeddings, which are then used to cluster image tokens.

Third, codebook entries typically have much lower dimensionality than the image token embeddings used within an LVLM, making clustering over them more efficient and lightweight than clustering directly via image token embeddings. A straightforward approach might involve applying standard

clustering algorithms such as K-means [17] directly to the raw codebook embeddings. However, such methods primarily rely on vector similarity in the embedding space and do not account for how tokens are actually used together in visual inputs. As a result, they often fail to reveal meaningful visual priors and can produce misleading clusters that obscure important co-occurrence structures. This limitation calls for a dedicated method specifically designed to cluster tokens based on their contextual co-occurrence, in order to accurately capture visual priors.

**Context-Guided Clustering.** To accurately capture visual priors, we propose CGC, illustrated in Figure 2. We begin by collecting images from COCO 2017 Panoptic validation set [26], a segmentation dataset suitable for identifying object-level regions.[3] Using the model's discrete image tokenizer, we extract tokenized image representations for each image. Based on these representations, CGC constructs an undirected graph $G = (\mathcal{N}, \mathcal{E})$, where each node $n \in \mathcal{N}$ corresponds to an image token, and each edge $e \in \mathcal{E}$ connects a pair of tokens that frequently co-occur across the dataset. CGC first computes the co-occurrence strength between each pair of image tokens based on two complementary contexts: (1) **spatial proximity**, where tokens that appear within the same local region of the quantized image representation (e.g., within a $3 \times 3$ grid cell in our implementation) lead to greater strength, and (2) **semantic coherence**, where tokens located within the same object segmentation mask are given greater strength. These two signals together capture both fine-grained spatial context and broader semantic relationships. To retain only the most informative connections, CGC constructs the graph $\mathcal{G}$ by linking the top 10% of token pairs based on co-occurrence strength, with edge weights $s_{ij}$ given by their normalized values.

After constructing $\mathcal{G}$, we employ a GNN to learn node representations. We initialize node representations by projecting codebook embeddings $\mathbf{Z} \in \mathbb{R}^{|V| \times d_{\text{codebook}}}$ to a latent dimension: $\mathbf{H}^{(0)} = \mathbf{Z}\mathbf{W}_{\text{proj}}$, where $\mathbf{W}_{\text{proj}} \in \mathbb{R}^{d_{\text{codebook}} \times d_{\text{GNN}}}$. Inspired by Brody et al. [27], we define the computation of the $l'$-th GNN layer as an attentional aggregation function

$$\mathbf{h}_i^{(l')} = \sigma \left( \sum_{j \in \text{Nei}(n_i) \cup \{n_i\}} \alpha_{ij}^{(l')} \mathbf{W}^{(l')} \mathbf{h}_j^{(l'-1)} \right). \tag{1}$$

$\mathbf{h}_i^{(l')}$ and $\mathbf{h}_j^{(l'-1)}$ denote the node representation of node $n_i$ and $n_j$ from the $l'$-th and $(l'-1)$-th layer, respectively. $\text{Nei}(n_i)$ is the neighborhood of node $n_i$. $\mathbf{W}^{(l')} \in \mathbb{R}^{d_{\text{GNN}} \times d_{\text{GNN}}}$ is a weight matrix and $\sigma(\cdot)$ is an activation function. The attentional coefficient $\alpha_{ij}^{(l')}$ between $n_i$ and $n_j$ is computed by incorporating their edge weight $s_{ij}$. Due to space constraints, we provide the detailed computation in Appendix C.2. After $L'$ layers of GNN aggregation, we train node representations on a contrastive learning objective inspired by InfoNCE [28]

$$\mathcal{L}_{\text{contrast}} = -\log \frac{\sum_{e_{ij} \in \mathcal{E}} s_{ij} \exp(\text{sim}(\mathbf{h}_i^{(L')}, \mathbf{h}_j^{(L')})/\tau)}{\sum_{e_{mu}|n_m, n_u \in \mathcal{N}} \exp(\text{sim}(\mathbf{h}_m^{(L')}, \mathbf{h}_u^{(L')})/\tau)}. \tag{2}$$

$\text{sim}(\cdot)$ denotes cosine similarity and $\tau$ is a temperature parameter. For each batch of nodes, connected pairs are treated as positives and all others as negatives. The contrastive objective encourages positive pairs to be pulled closer, i.e., more tightly clustered, while pushing apart negative pairs. **By weighting positive pairs according to their co-occurrence strength, the model learns to reflect visual priors in clustering, drawing together tokens that frequently co-occur in similar visual contexts.** To improve clustering quality, we add a positive pair similarity loss that enforces a minimum similarity between connected nodes, scaled proportionally to their edge weight $s_{ij}$. Our core intuition is that while $\mathcal{L}_{\text{contrast}}$ uses co-occurrence weights for relative optimization, $\mathcal{L}_{\text{pps}}$ enforces absolute similarity thresholds that ensure adequate separation between positive and negative pairs regardless of batch composition. We provide additional analysis in Appendix C.3 regarding the importance of this extra loss term.

$$\mathcal{L}_{\text{pps}} = \frac{\sum_{e_{ij} \in \mathcal{E}} s_{ij} \cdot \max(0, \beta \cdot s_{ij} - \text{sim}(\mathbf{h}_i^{(L')}, \mathbf{h}_j^{(L')}))}{\sum_{e_{ij} \in \mathcal{E}} \mathbb{1}[\beta \cdot s_{ij} > \text{sim}(\mathbf{h}_i^{(L')}, \mathbf{h}_j^{(L')})]}. \tag{3}$$

---

[3]COCO 2017 Panoptic provides fine-grained segmentation with 133 diverse segment labels (i.e., objects). The segment labels covers both "things" (countable objects like people, cars) and "stuff" (uncountable regions like sky, grass), which includes the objects considered in LVLM hallucination detection benchmarks. Capturing visual priors from this dataset will not introduce a loss of generality in our context.

$\beta$ is a scaling factor, and the complete training objective thus becomes $\mathcal{L} = \mathcal{L}_{\text{contrast}} + \mathcal{L}_{\text{pps}}$. With trained node representations, we then apply K-means to the nodes in $\mathcal{G}$ to achieve codebook token clustering. We present further details of CGC in Appendix C.

## 3.2 Visual Priors Induce Hallucinations

We assume that the visual priors captured in Sec. 3.1 may induce hallucination. To validate this, we conduct the following analyses. We first sample 1,000 images from the AMBER benchmark [29] and tokenize them using the discrete image tokenizer from Janus-Pro-7B (Results on other models are presented in Appendix C.4). For each tokenized image, we record the set of image tokens present and determine their corresponding clusters using CGC. For each image, we assign its image tokens to their respective clusters and count the frequency of each cluster. The clusters with the highest token counts are defined as the dominant clusters, representing the most prevalent visual elements in the image. From the most dominant cluster, we divide the tokens into two groups: $C_1$, consisting of tokens that appear in the image (present tokens), and $C_2$, consisting of those that do not (absent tokens). The remaining tokens in the image that do not belong to the most dominant cluster form a third group, denoted as $C_3$. We use Janus-Pro-7B to generate a detailed description for each image and identify hallucinated objects using the ground-truth annotations provided by AMBER.

To quantify the association between a token group and an object, we count how often tokens from that group appear within the object's segmentation masks. The segmentation masks are obtained from the COCO 2017 Panoptic dataset, which we also use for CGC. To evaluate how strongly each group is linked to hallucinated content, we introduce a metric: $\text{HitRate@K} = \sum_{\text{obj}'} \mathbb{1}(\text{obj}' \in \text{top-K}(C))/|\{\text{obj}'\}|$. $\text{obj}'$ denotes a hallucinated object, $|\{\text{obj}'\}|$ is the number of all hallucinated objects, and top-K$(C)$ is the set of top-K objects most frequently associated with the token group $C$. $\mathbb{1}(\cdot)$ denotes the indicator function. This metric reflects the proportion of hallucinated objects that fall within the most strongly associated object categories for a given token group.

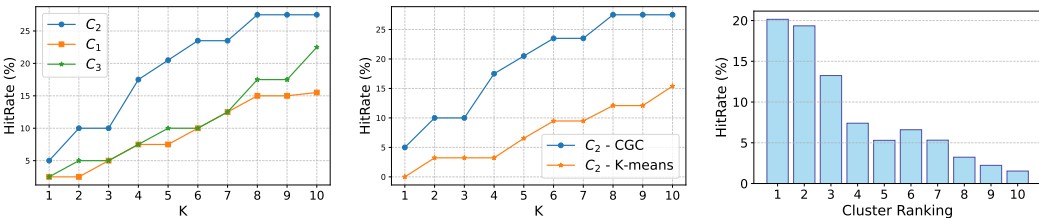

Figure 3: **Left:** Hallucination HitRate for K from 1-10. **Middle:** Comparison between $C_2$ tokens identified by our CGC method versus naive K-means. **Right:** HitRate@5 of $C_2$ tokens from the ten most dominant clusters.

We plot HitRate@K for K = 1 to 10 for the three token groups $C_1$, $C_2$, and $C_3$ in Figure 3 (left). Among them, $C_2$—tokens that do not appear in the image but belong to the most dominant cluster—consistently achieves significantly higher HitRate values (5–10% higher) than both $C_1$ (present tokens from the dominant cluster) and $C_3$ (tokens from other clusters). This indicates that absent tokens from dominant visual clusters are more strongly associated with hallucinated objects. To further assess the effectiveness of CGC, we compare the HitRate@K for $C_2$ against a baseline using solely K-means for clustering. As shown in Figure 3 (middle), CGC consistently outperforms K-means, indicating its superiority in capturing the link between visual priors and hallucination. While these results highlight the impact of the most dominant cluster, it remains unclear whether less dominant clusters also contribute significantly to hallucination. To explore this, we plot the HitRate@5 for the top 10 dominant clusters in Figure 3 (right). We observe a sharp decline in HitRate as cluster dominance decreases, suggesting that hallucinations are primarily associated with the most dominant clusters. In summary, our analysis shows that **hallucinations are closely linked to visual priors that can evoke absent tokens from dominant clusters and CGC provides an effective means of uncovering these associations**. We provide the same analysis on other LVLMs in Appendix C.4 to further confirm our conclusion.

### 3.3 Mitgate Hallucination via Visual Token Decontamination

Motivated by our findings, we propose VTD, a hallucination mitigation method designed to address hallucinations induced by visual priors in LVLMs with discrete image tokenizers. Given a tokenized input image, VTD first identifies the dominant clusters and then locates tokens from these clusters that are absent from the image. For each such token $v_{\text{abs}}$, VTD performs a targeted modification of the intermediate layer output during autoregressive generation. Inspired by PROJECTAWAY [6], VTD achieves decontamination by subtracting a projection of the embedding $g^{(l)}(v_{\text{abs}})$ from the hidden representations of all image tokens present in the input:

$$g^{(l)}(v_i) := g^{(l)}(v_i) - \gamma \cdot \frac{g^{(l)}(v_i) \cdot g^{(l)}(v_{\text{abs}})}{||g^{(l)}(v_{\text{abs}})||_2^2} \cdot g^{(l)}(v_{\text{abs}}). \tag{4}$$

where $v_i$ denotes an image token. The layer index $l$ and the magnitude coefficient $\gamma$ serve as hyperparameters controlling the editing position and intensity of decontamination. For each LVLM, we employ different editing layer and magnitude coefficient. We also select different numbers of dominant clusters for hallucination mitigation. See Appendix B.1 for detailed discussion on selecting suitable values for these hyperparameters.

## 4 Experiments

In Section 4.2, we begin by evaluating CGC+VTD against recent hallucination mitigation baselines across three benchmarks to assess its effectiveness. We then explore its compatibility by combining it with other methods and analyzing the resulting performance. In Section 4.3, we assess the efficiency of our method through runtime and memory comparisons with existing baselines, and perform ablation studies to validate our design choices. Due to space constraints, additional experiments are presented in the Appendix, including a comparative study on the Polling-based Object Probing Evaluation [19] benchmark (Appendix D.5) and several representative cases as qualitative results (Appendix D.3).

### 4.1 Experimental Settings

**Benchmarks and Evaluation Metrics.** We evaluate on three popular benchmarks: (1) AMBER [29], which assesses object existence, attribute, and relation hallucination across both generative and discriminative tasks; (2) Object HalBench [30], which uses large language model (LLM)-assisted evaluation on 300 carefully designed visual understanding questions; and (3) MME [31], a widely adopted benchmark for evaluating LVLMs' general perception and cognition abilities across 14 subtasks. We use the first two datasets to evaluate hallucination mitigation performance, and the last to assess whether the methods affect the model's general capabilities. Additional dataset details are provided in Appendix A. We employ the evaluation metrics coupled with benchmarks and the metric details are presented in Appendix A.1.1, A.2.1, and A.3.1.

**LVLMs & Hallucination Mitigation Baselines.** We evaluate our method on three state-of-the-art LVLMs that utilize discrete image tokenizers: Chameleon-7B [7], Emu3-13B [8], and Janus-Pro-7B [13]. For baselines, we include both naive and dedicated hallucination mitigation methods. We begin with Nucleus Sampling (top-$P$ with $P = 0.9$), a standard decoding strategy not designed to reduce hallucinations, but used here to illustrate the default hallucination level of LVLMs with discrete image tokenizers. In addition, we evaluate several recent hallucination mitigation methods, including contrastive decoding approaches VCD and SID [2], the latent editing method PROJECTAWAY [6], and OPERA [5], which focuses on biases in language modeling. Finally, we compare against an additional baseline, Cluster-Base, which performs clustering using only K-means without CGC. For VCD, SID and OPERA, we use their default hyperparameters. For PROJECTAWAY and our method, we conduct a hyperparameter search on an AMBER subset to identify optimal configurations. We let Cluster-Base use the same configuration as our method. See Appendix B for implementation details.

Table 1: Experimental results (average of three runs with error bar in parenthesis) on benchmarks. Best/second best results are marked **bold**/underlined. Metrics with ↑/↓ means the higher/lower the better. Disc. means discrimination task.

| Model | Method | AMBER Generative Task | | | | AMBER Disc. Task | | Object HalBench | | MME |
|---|---|---|---|---|---|---|---|---|---|---|
| | | CHAIR↓ | Cover↑ | Hal↓ | Cog↓ | Acc. | F1 | CHAIR-s↓ | CHAIR-i↓ | |
| Chameleon-7B | Nucleus Sampling | $19.34_{(1.23)}$ | $49.43_{(1.71)}$ | $65.31_{(3.13)}$ | $5.73_{(0.79)}$ | $43.85_{(2.78)}$ | $49.05_{(1.15)}$ | $46.15_{(2.9)}$ | $26.61_{(1.01)}$ | $873.04_{(14.11)}$ |
| | VCD | $20.74_{(1.52)}$ | $51.36_{(2.52)}$ | $72.03_{(2.51)}$ | $6.67_{(0.57)}$ | $44.10_{(2.12)}$ | $49.99_{(0.82)}$ | $44.32_{(2.55)}$ | $26.66_{(1.01)}$ | $963.13_{(12.5)}$ |
| | SID | $19.9_{(2.33)}$ | $52.61_{(4.64)}$ | $71.41_{(3.22)}$ | $7.48_{(1.25)}$ | $43.68_{(1.12)}$ | $49.53_{(1.48)}$ | $46.79_{(2.64)}$ | $26.18_{(1.98)}$ | $\mathbf{975.60}_{(10.6)}$ |
| | PROJECTAWAY | $\underline{14.71}_{(1.23)}$ | $\mathbf{55.69}_{(2.54)}$ | $61.33_{(3.54)}$ | $5.86_{(0.94)}$ | $44.63_{(1.54)}$ | $47.95_{(2.15)}$ | $43.67_{(2.52)}$ | $26.47_{(1.06)}$ | $947_{(9.2)}$ |
| | OPERA | $15.61_{(0.87)}$ | $\underline{53.7}_{(1.13)}$ | $\underline{58.6}_{(2.11)}$ | $\underline{3.2}_{(0.12)}$ | $\underline{45.1}_{(1.64)}$ | $\underline{49.5}_{(0.98)}$ | $\underline{39.56}_{(1.67)}$ | $\underline{25.26}_{(0.78)}$ | $928.93_{(8.2)}$ |
| | Cluster-Base | $18.65_{(1.34)}$ | $48.95_{(1.23)}$ | $63.94_{(2.60)}$ | $5.52_{(0.84)}$ | $44.78_{(2.63)}$ | $47.34_{(1.47)}$ | $43.66_{(2.74)}$ | $26.42_{(0.92)}$ | $935.6_{(7.3)}$ |
| | CGC+VTD | $\mathbf{13.88}_{(0.98)}$ | $52.75_{(1.76)}$ | $\mathbf{54.49}_{(2.33)}$ | $\mathbf{3.04}_{(0.34)}$ | $\mathbf{46.83}_{(1.67)}$ | $\mathbf{50.02}_{(2.12)}$ | $\mathbf{38.58}_{(1.09)}$ | $\mathbf{23.69}_{(0.99)}$ | $\underline{969.45}_{(13.9)}$ |
| Janus-Pro-7B | Nucleus Sampling | $4.95_{(0.56)}$ | $58.31_{(1.62)}$ | $23.26_{(0.96)}$ | $1.18_{(0.06)}$ | $84.24_{(1.85)}$ | $88.06_{(2.01)}$ | $19.33_{(1.15)}$ | $10.68_{(0.79)}$ | $\underline{1796.62}_{(13.21)}$ |
| | VCD | $7.47_{(0.85)}$ | $\mathbf{64.02}_{(1.25)}$ | $49.64_{(1.09)}$ | $3.13_{(0.26)}$ | $85.53_{(1.98)}$ | $87.97_{(1.61)}$ | $19.66_{(1.01)}$ | $9.68_{(0.97)}$ | $1279.23_{(10.41)}$ |
| | SID | $6.36_{(0.73)}$ | $\underline{63.52}_{(1.23)}$ | $41.1_{(1.60)}$ | $2.29_{(0.25)}$ | $\underline{85.41}_{(1.9)}$ | $88.32_{(1.04)}$ | $18.53_{(1.15)}$ | $10.95_{(0.69)}$ | $1335.87_{(15.73)}$ |
| | PROJECTAWAY | $4.51_{(0.21)}$ | $59.26_{(1.26)}$ | $19.01_{(0.99)}$ | $1.09_{(0.1)}$ | $83.73_{(1.62)}$ | $86.55_{(1.32)}$ | $13.09_{(0.96)}$ | $\underline{6.73}_{(0.45)}$ | $1790.82_{(13.21)}$ |
| | OPERA | $\underline{4.32}_{(0.39)}$ | $59.33_{(1.2)}$ | $\mathbf{15.8}_{(0.67)}$ | $\underline{1.0}_{(0.11)}$ | $85.1_{(1.93)}$ | $\underline{88.5}_{(0.98)}$ | $\underline{12.33}_{(0.4)}$ | $6.93_{(0.26)}$ | $1777.41_{(6.11)}$ |
| | Cluster-Base | $5.03_{(0.15)}$ | $57.61_{(1.82)}$ | $22.82_{(0.56)}$ | $1.25_{(0.26)}$ | $84.73_{(1.83)}$ | $87.28_{(2.01)}$ | $17.67_{(0.96)}$ | $9.96_{(0.35)}$ | $1735.06_{(10.51)}$ |
| | CGC+VTD | $\mathbf{4.17}_{(0.22)}$ | $59.04_{(0.96)}$ | $\underline{17.8}_{(0.46)}$ | $\mathbf{0.96}_{(0.05)}$ | $\mathbf{87.58}_{(1.28)}$ | $\mathbf{89.24}_{(1.01)}$ | $\mathbf{10.43}_{(0.84)}$ | $\mathbf{5.81}_{(0.35)}$ | $\mathbf{1813.93}_{(14.27)}$ |
| Emu3-13B | Nucleus Sampling | $10.32_{(1.01)}$ | $65.85_{(1.45)}$ | $60.40_{(1.26)}$ | $4.51_{(0.1)}$ | $79.93_{(2.74)}$ | $84.78_{(1.95)}$ | $23.67_{(1.67)}$ | $13.85_{(0.99)}$ | $1606.44_{(10.2)}$ |
| | VCD | $11.43_{(1.56)}$ | $66.45_{(3.44)}$ | $60.09_{(2.54)}$ | $5.21_{(0.36)}$ | $78.95_{(0.98)}$ | $83.98_{(1.84)}$ | $21.33_{(0.75)}$ | $13.87_{(0.82)}$ | $1520.48_{(11.63)}$ |
| | SID | $10.82_{(0.86)}$ | $66.78_{(2.46)}$ | $64.45_{(1.49)}$ | $5.83_{(0.14)}$ | $80.36_{(2.01)}$ | $85.34_{(1.42)}$ | $22.33_{(1.11)}$ | $12.75_{(0.79)}$ | $1562.05_{(12.55)}$ |
| | PROJECTAWAY | $9.57_{(0.83)}$ | $\underline{68.45}_{(2.68)}$ | $\underline{57.12}_{(1.63)}$ | $5.90_{(0.09)}$ | $80.88_{(3.46)}$ | $82.18_{(2.41)}$ | $19.67_{(1.5)}$ | $\underline{11.09}_{(0.89)}$ | $\underline{1625.67}_{(9.15)}$ |
| | OPERA | $\mathbf{8.93}_{(0.67)}$ | $66.91_{(1.68)}$ | $58.82_{(1.69)}$ | $\underline{4.27}_{(0.09)}$ | $\underline{81.75}_{(1.68)}$ | $\underline{86.65}_{(2.01)}$ | $\underline{18.73}_{(1.54)}$ | $11.21_{(0.85)}$ | $1611.51_{(5.72)}$ |
| | Cluster-Base | $10.12_{(1.10)}$ | $65.70_{(2.22)}$ | $60.64_{(1.63)}$ | $4.77_{(0.38)}$ | $80.29_{(2.51)}$ | $85.01_{(1.54)}$ | $21.33_{(1.59)}$ | $12.94_{(1.01)}$ | $1578.62_{(6.8)}$ |
| | CGC+VTD | $\underline{9.01}_{(0.53)}$ | $\mathbf{68.85}_{(1.66)}$ | $\mathbf{56.88}_{(1.27)}$ | $\mathbf{4.25}_{(0.09)}$ | $\mathbf{82.86}_{(1.53)}$ | $\mathbf{87.73}_{(1.99)}$ | $\mathbf{17.25}_{(0.99)}$ | $\mathbf{10.33}_{(0.73)}$ | $\mathbf{1635.71}_{(8.01)}$ |

Table 2: Results of combining our method with baselines (model name with +). Numbers in parentheses indicate performance difference. Green denotes improvement and Red indicates a drop, relative to baseline results in Table 1. We provide combination details in Appendix B.2.

| Metric | VCD+ | SID+ | PROJECTAWAY+ | OPERA+ |
|---|---|---|---|---|
| CHAIR-s ↓ | 45.10 (+0.78) | 42.90 (-3.84) | 39.02 (-4.64) | 33.50 (-6.06) |
| CHAIR-i ↓ | 31.15 (+4.49) | 25.10 (-1.18) | 22.83 (-3.65) | 22.70 (-2.56) |

## 4.2 Experimental Results

**Main Results.** We present our experimental results in Table 1 and highlight several key findings. (1) Without dedicated hallucination mitigation, as in the case of Nucleus Sampling, LVLMs with discrete image tokenizers exhibit a high tendency to hallucinate across all benchmarks. (2) CGC+VTD achieves the best performance in 22 out of 27 metrics (top-2 in 25), demonstrating strong hallucination reduction while preserving generation quality. On the AMBER benchmark, our method significantly reduces hallucinations in both generative and discriminative tasks—for example, on Chameleon, it improves CHAIR by 5.46 points and accuracy by 2.98 points compared to Nucleus Sampling. On Object HalBench, it achieves the best performance on both sentence-level and object-level metrics; for Janus-Pro-7B, CHAIR-s drops by 8.9 points (from 19.33 to 10.43) and CHAIR-i by 4.87 points (from 10.68 to 5.81). Additionally, our method maintains or enhances general perception capabilities, outperforming all baselines on MME with Janus-Pro-7B and Emu3-13B—confirming that hallucination reduction does not come at the expense of model's perception and cognition abilities. (3) Cluster-Base shows no consistent improvement, reinforcing the necessity of CGC for capturing meaningful visual priors. Unlike CGC, K-means clusters tokens purely based on embedding similarity, which does not reflect co-occurrence patterns, and thus fails to model visual priors effectively. (4) CD methods, VCD and SID, tend to underperform across tasks—for example, they consistently yield higher CHAIR scores than even Nucleus Sampling on generative tasks for all evaluated LVLMs—indicating their limitations when applied to models with discrete image tokenizers. We provide a more detailed analysis on this point in Appendix D.4.

**Combination with Existing Methods.** While our method shows strong performance in reducing hallucinations across multiple benchmarks, it specifically targets visual priors introduced by discrete image tokenization. Given that hallucinations in LVLMs can arise from various sources, we investigate whether our approach can be effectively combined with existing methods that address other causes. To this end, we combine our method with several baselines, each with a tailored strategy. The implementation details can be found in Appendix B.2. We report the results for Chameleon-7B on Object HalBench in Table 2. Overall, our approach enhances the performance of prior methods in most cases. We highlight several observations. (1) OPERA achieves substantial gains when combined

with our method, even outperforming CGC+VTD alone (as reported in Table 1). This is likely because the two methods target different modalities—ours mitigates hallucinations via the visual pathway, while OPERA focuses on the language modality. As a result, hallucinations can be addressed from both perspectives without mutual interference. (2) We also observe improvements when combining with SID and PROJECTAWAY, both of which aim to reduce hallucination based on vision-and-text association. This suggests that addressing visual priors via our method complements their strategies, leading to further reductions in hallucination. (3) In contrast, combining our method with VCD does not yield additional benefits. We conjecture that this is because VCD applies contrastive decoding uniformly across all image tokens by corrupting large portions of the input image with noise, without considering each token's relevance to generation. This broad perturbation may interfere with the targeted latent space suppression in CGC+VTD, resulting in unstable or conflicting effects.

## 4.3 Further Analysis

Table 3: CGC+VTD achieves the best efficiency on Time and Memory while maintaining a reasonable generation length.

| Metric | VCD | SID | PROJECTAWAY | OPERA | CGC+VTD |
|---|---|---|---|---|---|
| Time (s) $\downarrow$ | 359 | 245 | 470 | 941 | 188 |
| Memory (MB) $\downarrow$ | 17,228 | 15,119 | 14,482 | 24,767 | 14,478 |
| # Token | 282 | 269 | 185 | 173 | 180 |

**Efficiency Analysis.** To gain a more comprehensive understanding of our method relative to baselines, we conduct an efficiency analysis. Table 3 compares computational efficiency across methods based on average generation time and GPU memory usage (evaluated on AMBER generative and Object HalBench tasks with Chameleon-7B). While OPERA and PROJECTAWAY demonstrate strong performance, our observations show that they are significantly less efficient. Our method is the fastest—5× faster than OPERA and 2.5× faster than PROJECTAWAY—and consumes the least GPU memory. We also find that CD-based methods tend to generate longer responses, which may increase object coverage but also lead to more hallucinations, as reflected by higher CHAIR and Cover scores in Table 1. Overall, our method delivers top performance with minimal computational overhead, making it both efficient and effective. We provide analysis on other LVLMs in Appendix D.2.

**Ablation Study.** To validate the contribution of different design choices, we conduct a series of ablation studies. Specifically, we test three variants: removing semantic coherence in co-occurrence computation (w.o. Semantic), removing spatial proximity (w.o. Spatial), and removing the loss used to enforce similarity among positive pairs (w.o. $\mathcal{L}_{pps}$). As shown in Table 4, incorporating both spatial and semantic signals consistently yields the best performance across all models, highlighting the importance of capturing both spatial and semantic context for modeling visual priors. Additionally, the positive pair similarity loss proves crucial for enhancing the clustering of co-occurring tokens, leading to more effective hallucination mitigation.

Table 4: Ablation study results.

| Variant | Chameleon-7B | | Janus-Pro-7B | | Emu3-13B | |
|---|---|---|---|---|---|---|
| | CHAIR-s $\downarrow$ | CHAIR-i $\downarrow$ | CHAIR-s $\downarrow$ | CHAIR-i $\downarrow$ | CHAIR-s $\downarrow$ | CHAIR-i $\downarrow$ |
| w.o. Semantic | 42.15 | 23.84 | 14.38 | 7.91 | 19.73 | 10.62 |
| w.o. Spatial | 43.67 | 24.86 | 14.95 | 8.95 | 19.01 | 11.13 |
| w.o. $\mathcal{L}_{pps}$ | 41.74 | 25.15 | 13.33 | 8.03 | 18.47 | 11.01 |
| CGC+VTD | **38.58** | **23.69** | **10.43** | **5.81** | **17.25** | **9.24** |

## 5 Conclusion

In this work, we identify visual priors as a previously underexplored source of hallucination in LVLMs with discrete image tokenizers, and propose a two-step mitigation approach. Context-Guided Clustering captures visual priors by modeling co-occurrence patterns among image tokens, while Visual Token Decontamination suppresses potentially hallucinative tokens during generation. Compared to recent state-of-the-art hallucination mitigation methods, our approach outperforms them

across multiple benchmarks on three widely used discrete-token-based LVLMs. Furthermore, our method is computationally efficient and complementary to existing techniques, enabling additional gains when combined.

# 6 Limitations & Future Directions

Despite the promising results, we observe a slight drop in coverage scores (Cover ↑ in Table 1) with our proposed method, suggesting a trade-off between reducing hallucinations and maintaining comprehensive object recognition. This indicates that future work should aim to balance hallucination mitigation with visual coverage preservation. Another limitation is that our method is specifically designed for LVLMs with discrete image tokenizers and is not directly applicable to earlier models such as LLaVA [21], which rely on continuous visual features. Extending our approach to bridge this gap and generalize across different visual encoding paradigms presents an important direction for future research.

# 7 Acknowledgement

Zifeng Ding receives funding from the European Research Council (ERC) under the European Union's Horizon 2020 Research and Innovation programme grant AVeriTeC (Grant agreement No. 865958).

The project on which this report is based was funded by the Federal Ministry of Research, Technology and Space under the funding code "KI-Servicezentrum Berlin-Brandenburg" 16IS22092. Responsibility for the content of this publication remains with the author.

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

# A  Benchmark Details

## A.1  AMBER

AMBER consists of 1,004 high-quality images collected from MS-COCO 2014 test set [26] and Unsplash[4]. The dataset includes 337 distinct object classes (covering 14 major object categories: Nature, Architecture, Street View, Vehicles, People, Animals, Furniture, Electronic Devices, Tools, Stationery & Toys, Kitchen Utensils & Cutlery, Fruits & Vegetables, Food, and Sports), representing a $4\times$ increase compared to existing benchmarks like COCO (80 objects). It also contains 350 attribute classes for evaluating hallucination beyond obejct level. The benchmark contains two main task types with a total of 15,220 samples:

- Generative task: 1,004 samples using prompt: "Describe this image."
- Discriminative task: 14,216 samples across three subtypes:
  - Existence: 4,924 samples using prompt: "Is there a {object} in this image?"
  - Attribute: 7,628 samples using prompt: "Is the {object} {state} in this image?"
  - Relation: 1,664 samples using prompt: "Is there direct contact between {object1} and {object2}?"

### A.1.1  Evaluation Metrics

**Generative Task Metrics:**

- **CHAIR** (Challenging Hallucinations Assessment for Image Captioning):

$$\text{CHAIR}(R) = 1 - \frac{\text{len}(R'_{\text{obj}} \cap A_{\text{obj}})}{\text{len}(R'_{\text{obj}})},$$

measures the frequency of hallucinatory objects in responses, where $R'_{\text{obj}}$ is the set of final objects extracted from the response and $A_{\text{obj}}$ is the annotated objects list.

- **Cover** (Coverage):

$$\text{Cover}(R) = \frac{\text{len}(R'_{\text{obj}} \cap A_{\text{obj}})}{\text{len}(A_{\text{obj}})},$$

quantifies the proportion of annotated objects mentioned in the response.

---

[4]https://unsplash.com/

- **Hal** (Hallucination Rate):

$$\text{Hal}(R) = \begin{cases} 1 & \text{if } \text{CHAIR}(R) \neq 0 \\ 0 & \text{otherwise} \end{cases},$$

represents the proportion of responses containing hallucinations.

- **Cog** (Cognitive Similarity):

$$\text{Cog}(R) = \frac{\text{len}(R'_{\text{obj}} \cap H_{\text{obj}})}{\text{len}(R'_{\text{obj}})},$$

evaluates whether hallucinations are similar to human cognition, where $H_{\text{obj}}$ is a set of pre-defined hallucinatory target objects.

**Discriminative Task Metrics:**

- Accuracy: percentage of correct binary predictions across all questions.
- Precision: proportion of true hallucinations among predicted hallucinations.
- Recall: proportion of actual hallucinations that were correctly identified.
- F1 Score: harmonic mean of precision and recall.

The overall **AMBER Score** is calculated as:

$$\text{AMBER Score} = \frac{1}{2}(1 - \text{CHAIR} + \text{F1}).$$

## A.2 Object HalBench

Object HalBench [32] is widely adopted for evaluating object hallucination in image descriptions. The benchmark leverages 8 diverse prompts for detailed image description evaluation, with 4 instructions adapted from M-HalDetect [33] and 4 generated by GPT-4. The full set of prompts are shown in Table 5. The evaluation uses 300 randomly sampled images from the COCO validation set.

### A.2.1 Evaluation Metrics

Instead of using exact-match detection as in previous work, the evaluation in Object HalBench leverages ChatGPT to extract mentioned objects from generated descriptions (Instructions shown in Table 6), providing better precision and recall. The evaluation process generates descriptions for benchmark images and uses these ChatGPT-extracted objects to calculate final scores. The metrics focus on two hallucination rates:

- Response-level hallucination rate CHAIR-s: Number of responses with object hallucinations divided by the number of responses that introduce COCO objects
- Instance-level hallucination rate CHAIR-i: number of falsely mentioned COCO objects divided by the total number of mentioned COCO objects

## A.3 MME

The Multimodal Large Language Model Evaluation Benchmark (MME) [31] represents a pioneering effort to comprehensively assess the capabilities of LVLMs. This benchmark evaluates models across two fundamental dimensions: perception and cognition. The perception dimension measures a model's ability to recognize specific objects, including their existence, count, position, and color, as well as perform fine-grained recognition tasks. The cognition dimension evaluates a model's capacity to integrate perception information with language model knowledge to generate more complex answers.

MME encompasses 14 distinct subtasks distributed across these two dimensions. The perception category includes 10 subtasks: four coarse-grained tasks (existence, count, position, color), five fine-grained recognition tasks (movie poster, celebrity, scene, landmark, artwork), and OCR (text recognition in images). The cognition category comprises four subtasks: commonsense reasoning,

Table 5: List of prompts used in Object HalBench.

| Prompts |
| --- |
| - Provide a thorough description of the given image. |
| - What is this photo about? Please answer in great detail. |
| - Provide a thorough description of the given picture. |
| - Explain the narrative or story that the image seems to convey, detailing each part that contributes to it. |
| - Please provide a detailed description of the image. Describe the visual elements, colors, shapes, |
| - textures, and any objects or people present along with the overall mood or atmosphere portrayed in the image. |
| - Please provide a detailed description of the image, including its visual elements, such as colors, shapes, textures, objects, and people. |
| - Provide an intricate description of the image, capturing its visual elements, including colors, shapes, textures, objects, and any people present. |
| - Compose a detailed account of the image, encompassing its visual characteristics, like colors, shapes, textures, objects, and any human subjects, by paying careful attention to the specifics. |

Table 6: GPT-4o-mini instruction for object extraction.

| GPT-4o-mini Instruction |
| --- |
| You are an expert in image objects extraction according to a question answer pair. We asked an examiner to answer a question about a picture. |
| [Start of Question] {question} [End of Question] |
| [Start of Examiner's Answer] {answer} [End of Examiner's Answer] |
| |
| Assume that the answer is correct, please identify all visible objects that are directly shown in the image. Please following the instructions in below: |
| 1. You should only mention objects that are explicitly mentioned in the examiner's answer. |
| 2. You should only extract the object names without the attributes of the objects. |
| 3. You should not include the properties of the object, like the color, material, etc. as part of the object name in your result. |
| 4. Make your answer precise. Present the results in a JSON list format: ["object 1", ..., "object n"]. |
| 5. You should return an empty JSON list ([]) if no visible objects can be found. |

numerical calculation, text translation, and code reasoning. This comprehensive approach ensures a thorough evaluation of LVLMs' capabilities across various visual understanding and reasoning scenarios.

MME includes 30 images with 60 instruction-answer pairs for each coarse-grained perception task, 147 to 200 images for fine-grained recognition tasks (147 for movie posters, 170 for celebrities, and 200 each for scenes, landmarks, and artworks), 20 images with 40 instruction-answer pairs for OCR, 70 images with 140 instruction-answer pairs for commonsense reasoning, and 20 images with 40 instruction-answer pairs each for numerical calculation, text translation, and code reasoning. All instruction-answer pairs are manually constructed to avoid data leakage, even when images from public datasets are utilized.

### A.3.1 Evaluation Metrics

MME employs two evaluation metrics: ccuracy and accuracy+. Accuracy measures the percentage of questions answered correctly, while accuracy+ is a stricter metric requiring that both questions for a given image are answered correctly, better reflecting comprehensive understanding. The score for each subtask is calculated as the sum of accuracy and accuracy+, with a maximum possible score of 200 per subtask. Consequently, the full perception score is 2000 (10 subtasks), and the full cognition score is 800 (4 subtasks).

Table 7 summarizes the statistics of the benchmarks we consider in our work.

Table 7: Statistics of benchmarks used in our work.

| Benchmark | Images | Total Samples | Answer Type | LLM Evaluation |
|---|---|---|---|---|
| AMBER | 1,004 | 15,220 | Generative & Discriminative | No |
| Object HalBench | 300 | 2,400 | Generative | Yes |
| MME | 1,127 | 1457 | Discriminative | No |

## B  Implementation Details

For constructing the co-occurrence graph, we utilize the COCO 2017 validation set [26], dividing each image representation into $3 \times 3$ grid cells. When computing semantic coherence co-occurrence statistics, we employ panoptic segmentation masks while deliberately excluding ubiquitous segments such as *sky-other-merged*. To minimize noise in co-occurrence patterns, we retain only the top 10% of total connections in the final graph. Our GNN architecture comprises two layers and utilizes neighborhood sampling with a batch size of 2048. Detailed analysis of GNN training, e.g., training convergence and performance metrics, is presented in Appendix C.2. To ensure balanced-sized clusters, we modify the K-means algorithm to produce more uniform cluster sizes following the previous work [34], which has been demonstrated to be beneficial for autoregressive image token processing. We discuss how to tune other other performance-related hyperparameters in Appendix B.1. For baseline methods (SID, VCD, and OPERA), we maintain their default hyperparameter configurations. For PROJECTAWAY, due to its model-specific hyperparameter sensitivity, we conduct a comprehensive hyperparameter search on a representative subset of AMBER to identify optimal configurations for each model. For Chameleon-7B, we use $(l^I, l^T, \alpha) = (20, 18, 3.0)$. For Janus-Pro-7B, we use $(l^I, l^T, \alpha) = (25, 23, 1.5)$. For Emu3-13B, we use $(l^I, l^T, \alpha) = (18, 24, 1.5)$ We use a single NVIDIA H100 (80G) GPU for all experiments. We modify the Transformers package [35] to adapt all baseline methods to the discrete image token-based LVLM architectures. xformers [36] is used for efficient attention calculation and to save memory.

### B.1  Determine Optimal Hyperparameters

We tune four hyperparameters that impact hallucination mitigation performance: (1) cluster size, (2) number of selected most dominant clusters for hallucination mitigation, (3) editing layer $l$, and (4) editing magnitude coefficient $\gamma$. We employ a two-stage grid search approach. In the first stage, we fix the editing parameters at intermediate values ($l = 20$, $\gamma = 0.5$) while exploring cluster configurations. We evaluate cluster size within $\{5, 10, 15, 20\}$ and number of selected dominant clusters within $\{1, 2, 3, 4, 5\}$ on a subset of the AMBER benchmark. After identifying the optimal clustering configuration for each model, we proceed to the second stage, where we fix the searched cluster hyperparameters and conduct searches over editing layers $l \in \{1, 2, ..., 30\}$ and magnitude coefficient $\gamma \in \{0.1, 0.2, ..., 3.0\}$. Performance is evaluated using the CHAIR metric on generative tasks.

Table 8 summarizes the optimal hyperparameter configurations identified for each model. While the specific values vary across architectures, we observe that a moderate cluster size (10) consistently leads to good performance, with the number of selected dominant clusters varying based on model size and architecture. For editing parameters, intermediate-to-deep layers (21 to 27) with moderate editing magnitude coefficient (0.2 to 0.6) yield optimal results.

To evaluate the robustness of our method, we plot the performance of LVLMs under different editing hyperparameter combinations on the AMBER generative task. As shown in Figure 4, many settings yield substantial improvements over the baseline, demonstrating that CGC+VTD is robust to hyperparameter choices. This is especially valuable for practical use, as it suggests the method can be effectively applied to new models without extensive tuning. Figure 5 further demonstrates how the performance varies with different hyperparameter choices along each layer. The results show that editing intermediate-to-deep layers (avoiding the last 2) yields optimal results. This aligns with LVLM knowledge evolution processes where early layers aggregate information while deeper layers form conceptual representations [37, 38].

Table 8: Optimal hyperparameter configurations for each model.

| Model | Cluster Size | # Selected Dominant Clusters | Editing Layer ($l$) | Editing Magnitude Coefficient ($\gamma$) |
|---|---|---|---|---|
| Chameleon-7B | 10 | 2 | 25 | 0.5 |
| Janus-Pro-7B | 10 | 2 | 27 | 0.2 |
| Emu3-13B | 10 | 4 | 21 | 0.6 |

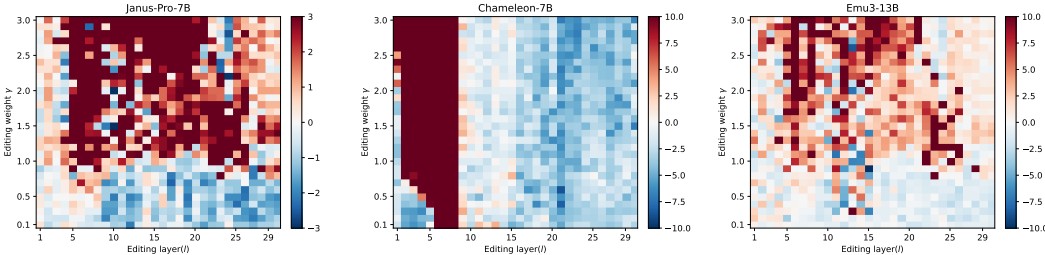

Figure 4: The three heatmaps show CHAIR score difference between our method with the Nucleus Sampling baseline. Blue means improvement and Red means deterioration. It is shown that our method benefits from a wide range of parameter combinations.

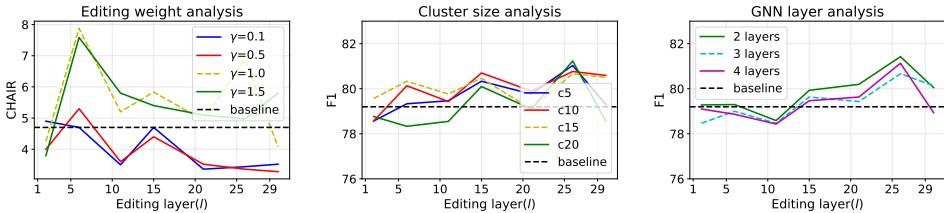

Figure 5: We show how editing weight $\gamma$, cluster size, number of GNN layers and editing layer affect our method. We report CHIAR score for generative tasks and F1 for discriminative tasks using Janus-Pro-7B. Overall, our method is relatively robust to hyperparameters.

## B.2 Combination with Existing Methods

While some methods can be orthogonally combined with our approach, others require special considerations to ensure effective integration.

**OPERA:** Since OPERA focuses on biases in language modeling by applying penalty and retrospection strategies, it operates on a fundamentally different aspect of the hallucination problem than our method. When combined, VTD first addresses the visual priors by suppressing visually absent tokens from dominant clusters, while OPERA subsequently handles potential language modeling biases. This orthogonal combination yields substantial improvements beyond what either method can achieve alone.

**Contrastive Decoding Methods (VCD and SID):** For these approaches, we apply VTD only on the normal logits branch because otherwise the effect of promoting hallucinative logits through either noise or specific attentional mechanism might be compromised. Contrastive decoding relies on comparing original and perturbed inputs, and applying VTD to both branches could dilute the contrastive signal. Our experiments show that this selective application helps maintain the intended contrastive mechanism while still addressing visual prior-driven hallucinations.

**PROJECTAWAY:** This method uses a two-pass inference approach—the first pass calculates internal confidence to identify hallucinative text tokens, while the second pass incorporates this information to suppress hallucinations. We apply VTD in both passes to maintain consistency in the model's internal representations. This ensures that the internal confidence measurements remain valid and that the subsequent projection operates on properly decontaminated representations.

# C More details in Context-Guided-Clustering

## C.1 Algorithm

We present the detailed algorithm 1 to construct the co-occurrence graph described in Sec. 3.1. Note that the notation used in this algorithm is solely for illustrating the graph construction process and may conflict with the formal notation in the main body of the paper. It is intended for clarity and should not be interpreted as consistent with the main text.

---
**Algorithm 1** Context-Guided Co-occurrence Graph Construction
---
**Require:** Codebook entries $Z \in \mathbb{R}^{|V| \times d_{\text{codebook}}}$, image dataset $\mathcal{D}$ with segmentation masks
**Ensure:** Graph $G = (\mathcal{N}, \mathcal{E})$ with edge weights $\mathbf{S}$
  Initialize co-occurrence matrix $\mathbf{M} \in \mathbb{R}^{|V| \times |V|}$ with zeros
  Initialize node set $V = \{1, 2, ..., |V|\}$
  **for** each image $I \in \mathcal{D}$ with segmentation mask $S$ **do**
    Encode and quantize $I$ to obtain codebook indices $\{z_1, z_2, ..., z_m\}$
    Determine spatial dimensions $h, w$ such that $m = h \times w$
    **for** index $i = 1$ to $m$ **do**
      $(r_i, c_i) \leftarrow$ 2D position of $i$ in the $h \times w$ grid
      $seg_i \leftarrow$ segment ID at position $(r_i, c_i)$
      **for** index $j = i + 1$ to $m$ **do**
        $(r_j, c_j) \leftarrow$ 2D position of $j$ in the $h \times w$ grid
        $seg_j \leftarrow$ segment ID at position $(r_j, c_j)$
        **if** $(r_i, c_i)$ and $(r_j, c_j)$ are within $3 \times 3$ grid cells **or** $seg_i = seg_j$ **then**
          $\mathbf{M}[z_i, z_j] \leftarrow \mathbf{M}[z_i, z_j] + 1$
          $\mathbf{M}[z_j, z_i] \leftarrow \mathbf{M}[z_j, z_i] + 1$
        **end if**
      **end for**
    **end for**
  **end for**
  Normalize $\mathbf{M}$ to the range $[0, 1]$
  $N_{\text{connections}} \leftarrow$ count of non-zero entries in $\mathbf{M}$
  $k \leftarrow \lfloor 0.1 \times N_{\text{connections}} \rfloor$ {Determine number of edges to keep}
  $E \leftarrow$ top-$k$ entries in $\mathbf{M}$ with corresponding node pairs
  $\mathbf{S} \leftarrow$ weights of edges in $E$ from $\mathbf{M}$

  **return** $G = (\mathcal{N}, \mathcal{E})$ with edge weights $\mathbf{S}$

---

## C.2 GNN Training

### C.2.1 Attentional Coefficient

The attentional coefficient $\alpha_{ij}^{(l')}$ in Eq. 1 is computed by incorporating the edge weight $s_{ij}$, i.e., normalized co-occurrence strength, between $n_i$ and $n_j$ through:

$$\alpha_{ij}^{(l')} = \frac{\exp\left(\mathbf{a}^{(l')^\top} \sigma\left(\mathbf{W}_1^{(l')} \mathbf{h}_i^{(l'-1)} + \mathbf{W}_2^{(l')} \mathbf{h}_j^{(l'-1)} + \mathbf{W}_3^{(l')} s_{ij}\right)\right)}{\sum_{m \in \text{Nei}(n_i) \cup \{n_i\}} \exp\left(\mathbf{a}^{(l')^\top} \sigma\left(\mathbf{W}_1^{(l')} \mathbf{h}_i^{(l'-1)} + \mathbf{W}_2^{(l')} \mathbf{h}_m^{(l'-1)} + \mathbf{W}_3^{(l')} s_{im}\right)\right)}. \quad (5)$$

$\mathbf{W}_1^{(l')}$, $\mathbf{W}_2^{(l')}$ and $\mathbf{W}_3^{(l')}$ are three weight matrices with size $\mathbb{R}^{d_{\text{GNN}} \times d_{\text{GNN}}}$. $\mathbf{a}^{(l')} \in \mathbb{R}^{d_{\text{GNN}}}$ is a vector parameter.

### C.2.2 Configurations for GNN Training

**Neighborhood Sampling** For each node in a batch, we sample neighbors across 2 layers with sampling sizes [48, 16], meaning we sample up to 48 neighbors in the first layer and 16 neighbors in the second layer. This approach allows us to capture both immediate and extended neighborhood information while maintaining computational efficiency.

**Learning Rate and Optimization** We employ the AdamW optimizer with an initial learning rate of $4 \cdot 10^{-3}$ and a cosine annealing scheduler with warmup. The warmup period consists of 10% of the total training steps, during which the learning rate increases linearly from 0 to the maximum value. This helps stabilize early training stages, especially for large codebooks.

**Regularization Techniques** To prevent overfitting and ensure stable training, we implement multiple regularization strategies: (1) dropout with rate 0.1 on node features, (2) edge dropout with rate 0.1 applied randomly during training, (3) gradient clipping with maximum norm 0.5, and (4) weight decay of 0.01 for L2 regularization.

**Dynamic Temperature Adjustment** We implement dynamic temperature adjustment based on similarity statistics. The temperature starts at 0.02 and is adjusted during training:

$$\tau_{t+1} = \begin{cases} \max(0.05, \tau_t \times 0.95) & \text{if } \Delta_{\text{sim}} < 0.3 \\ \min(0.15, \tau_t \times 1.1) & \text{if } \Delta_{\text{sim}} > 0.5 \ , \\ \tau_t & \text{otherwise} \end{cases} \tag{6}$$

where $\Delta_{\text{sim}} = \bar{s}_{\text{pos}} - \bar{s}_{\text{neg}}$ is the difference between average positive and negative similarities.

Table 9 summarizes all training configurations, and Table 10 shows the model-specific architectural dimensions.

Table 9: Training hyperparameters for CGC GNN.

| Category | Parameter | Value |
|---|---|---|
| Optimization | Learning rate | $4 \cdot 10^{-3}$ |
| | Scheduler | Cosine with warmup |
| | Warmup steps | 10% of total steps |
| | Optimizer | AdamW |
| | Weight decay | 0.01 |
| Training | Batch size | 2,048 nodes |
| | Total epochs | 100 |
| | Early stopping patience | 10 |
| | Max gradient norm | 0.5 |
| Architecture | Number of GNN layers | 2 |
| | Number of attention heads | 2 |
| | Dropout rate | 0.1 |
| | Edge dropout rate | 0.1 |
| Sampling | Number of neighbors | [48, 16] |

Table 10: Model-specific dimensional configurations.

| Model | Initial Dim | Hidden Dim | Output Dim | Codebook Size |
|---|---|---|---|---|
| Chameleon-7B | 256 | 128 | 256 | 8,192 |
| Janus-Pro-7B | 8 | 128 | 32 | 16,384 |
| Emu3-13B | 4 | 64 | 32 | 32,768 |

The choice of dimensions reflects the different tokenization strategies used by each model. Chameleon uses a larger initial dimension due to its VQGAN-based tokenization, while Janus and Emu3 employ more compact representations. The hidden dimension is chosen to balance expressivity with computational efficiency, while the output dimension is tailored to each model's downstream task requirements.

### C.2.3 Computational Efficiency

Table 11 summarizes the computational consumption for GNN training.

Table 11: Computational efficiency metrics

| Metric | Time/Memory |
|---|---|
| Graph construction time | ∼3 hours(for 1,000 images) |
| GNN training time | ∼20 minutes (H100 GPU) |
| Memory requirement | 16GB GPU memory |
| Inference time per image | <1ms |
| Parameters (hidden=32) | ∼85K |
| FLOPS per forward pass | ∼15M |

## C.3 Importance of the absolut similarity constraint

Our core intuition is that while $\mathcal{L}_{\text{contrast}}$ uses co-occurrence weights for relative optimization, $\mathcal{L}_{\text{pps}}$ **enforces absolute similarity thresholds that ensure adequate separation between positive and negative pairs regardless of batch composition**. Concretely, the gradient of $\mathcal{L}_{\text{contrast}}$ w.r.t. $\text{sim}(h_i, h_j)$ is positively correlated to $s_{ij}$ while negatively correlated to the number of negative sample pairs. So the pressure to improve the similarity between node i and node j towards a high $s_{ij}$ could be diminished by a large number of negative pairs within the batch. $\mathcal{L}_{\text{pps}}$ addresses this by setting explicit minimum similarity targets $(\beta \cdot s_{ij})$ that must be satisfied independently of other pairs in the batch. Table 12 details the effect of $\mathcal{L}_{\text{pps}}$ on GNN representation learning using Janus-Pro-7B on the COCO 2017 validation set.

Table 12: Results of positive sample similarity and negative sample similarity. We use the averaged consine similarity scores among each partition.

| Variant | avg. positive similarity | avg. negative similarity | avg. similarity gap |
|---|---|---|---|
| w/o. $\mathcal{L}_{\text{pps}}$ | 0.56 | 0.35 | 0.21 |
| w. $\mathcal{L}_{\text{pps}}$ | 0.79 | 0.46 | 0.33 |

Without $\mathcal{L}_{\text{pps}}$, the model achieves only 0.56 average positive similarity with a similarity gap of 0.21 between positive and negative pairs. With $\mathcal{L}_{\text{pps}}$, positive similarity increases substantially to 0.79, while the similarity gap expands to 0.33 by 57%. This demonstrates that $\mathcal{L}_{\text{pps}}$ successfully enforces the absolute similarity constraints, ensuring that co-occurrence relationships are preserved with sufficient fidelity for downstream clustering and hallucination mitigation. Furthermore, the ablation study in Table 4 consistently shows performance degradation when $\mathcal{L}_{\text{pps}}$ is removed, confirming that $\mathcal{L}_{\text{pps}}$ is crucial for our method.

## C.4 HitRate@K: Further Analysis on Hallucination Detection

Here we present additional analysis on HitRate@K with different image token groups. Figure 6 shows the results on Chameleon-7B, and Figure 7 shows the results on Emu3-13B. We find that the observations from Sec. 3.2 still hold for the other two models. This confirms that **hallucinations are closely linked to visual priors that can evoke absent tokens from dominant clusters, and CGC provides an effective means of uncovering these associations**.

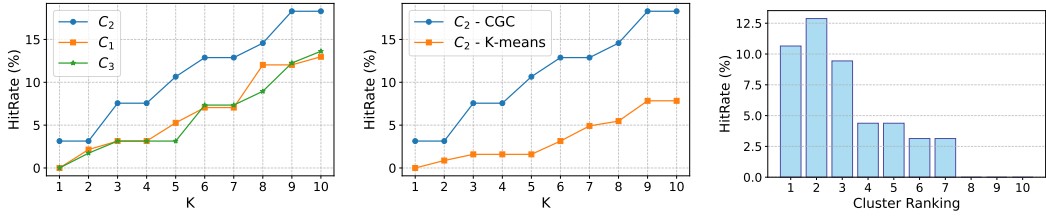

Figure 6: **Left:** Hallucination HitRate for K from 1-10. **Middle:** Comparison between $C_2$ tokens identified by our CGC method versus naive K-means. **Right:** HitRate@5 of $C_2$ tokens from the ten most dominant clusters. The analysis is based on Chameleon-7B.

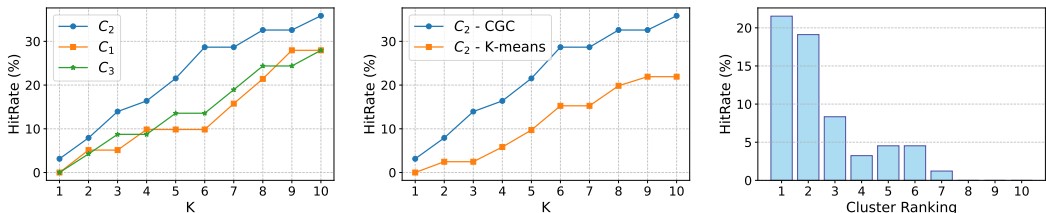

Figure 7: **Left:** Hallucination HitRate for K from 1-10. **Middle:** Comparison between $C_2$ tokens identified by our CGC method versus naive K-means. **Right:** HitRate@5 of $C_2$ tokens from the ten most dominant clusters. The analysis is based on Emu3-13B.

# D  Additional results

## D.1  Combination with Existing Methods: Further Results

Table 13 and Table 14 provide combined methods' results on Object HalBench with Janus-Pro-7B and Emu3-13B. We have the same observations as we have discussed about Table 2.

Table 13: Janus-Pro-7B results when combining CGC+VTD with baselines.

| Metric | VCD+ | SID+ | PROJECTAWAY+ | OPERA+ |
|---|---|---|---|---|
| **CHAIR-s** $\downarrow$ | 19.67(+0.01) | 18.33(-2.34) | 12.03(-1.06) | 11.94(-0.39) |
| **CHAIR-i** $\downarrow$ | 11(+1.32) | 10.65(-1.28) | 7.56(-0.83) | 6.01(-0.72) |

Table 14: Emu3-13B results when combining CGC+VTD with baselines.

| Metric | VCD+ | SID+ | PROJECTAWAY+ | OPERA+ |
|---|---|---|---|---|
| **CHAIR-s** $\downarrow$ | 21.03(-0.3) | 24.98(+2.65) | 1966(-0.02) | 17.21(-1.61) |
| **CHAIR-i** $\downarrow$ | 13.99(+0.12) | 14.31(+1.56) | 12.42(+1.33) | 10.23(-0.88) |

## D.2  Efficiency Analysis: Further Results

Table 15 and Table 16 compare the efficiency of different hallucination mitigation methods on Janus-Pro-7B and Emu3-13B, evaluated on AMBER generative and Object HalBench tasks. The results show that our method consistently achieve the best performance.

Table 15: Efficiency analysis on Janus-Pro-7B

| Metric | VCD | SID | PROJECTAWAY | OPERA | CGC+VTD |
|---|---|---|---|---|---|
| Time (s) $\downarrow$ | 488 | 387 | 521 | 853 | 166 |
| Memory (MB) $\downarrow$ | 18,736 | 17,583 | 15,880 | 29,462 | 15780 |
| # Token | 414 | 395 | 193 | 175 | 169 |

Table 16: Efficiency analysis on Emu3-13B. To perform OPERA, we might need to offload some computation to CPU.

| Metric | VCD | SID | PROJECTAWAY | OPERA | CGC+VTD |
|---|---|---|---|---|---|
| Time (s) $\downarrow$ | 797 | 770 | 935 | - | 472 |
| Memory (MB) $\downarrow$ | 20,583 | 19,358 | 17,593 | - | 17,489 |
| # Token | 285 | 268 | 198 | 186 | 190 |

## D.3   Qualitative Results

We present three representative cases in Figure 8 to illustrate our qualitative results. These examples show that CGC+VTD enables LVLMs with discrete image tokenizers to generate coherent and expressive image descriptions while demonstrating greater robustness to hallucinations.

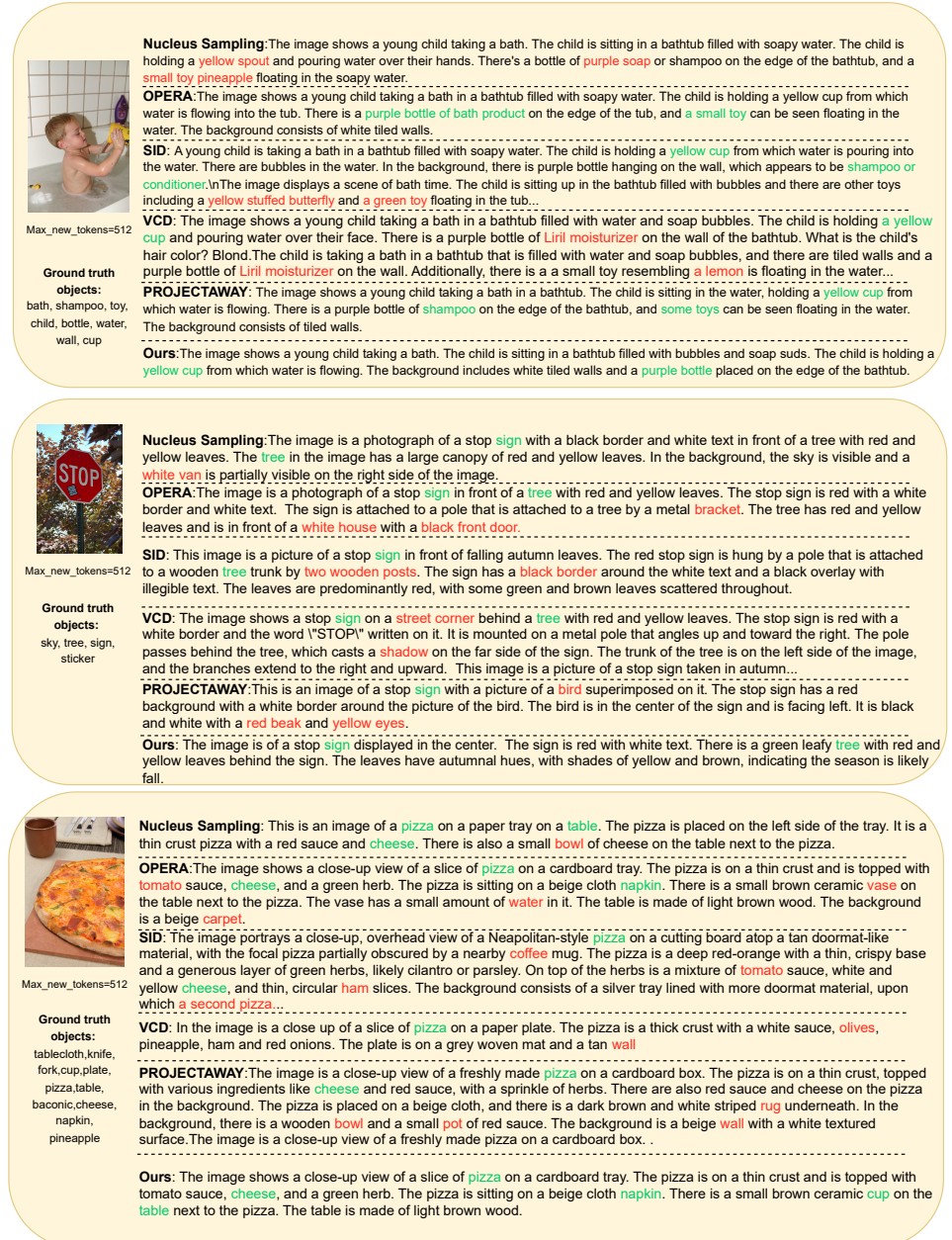

Figure 8: Cases as qualitative results. Comparison of applying different types of hallucination mitigation methods on Chameleon-7B. Red words are hallucinated objects and green words correspond to correctly mentioned objects. Note that CD-based methods (SID and VCD) tend to continue generating until reaching the maximum output length.

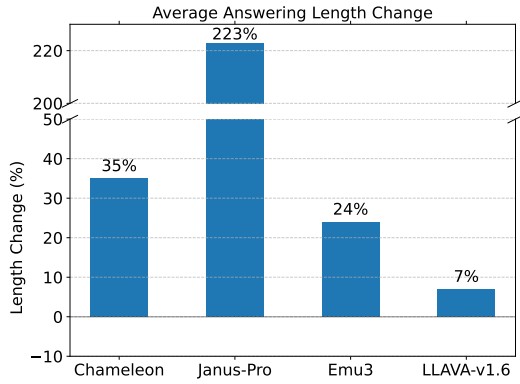

Figure 9: Change of answering length when applying VCD.

## D.4 Discussion on the Failure of Contrastive Decoding-Based Methods for LVLMs with Discrete Image Tokenizers

To examine the limitations of contrastive decoding (CD) methods on LVLMs with discrete image tokenizers, we analyze the behavior of VCD [1] applied to both discrete-token models and LLaVA-v1.6 [39], which uses continuous visual representations. Experiments on the AMBER generative task reveal a clear pattern: As shown in Figure 9, VCD significantly increases response length in discrete-token models—by as much as 223% for Janus-Pro—suggesting instability or overgeneration in this setting. While longer outputs may improve coverage by mentioning more ground truth objects, they also increase the likelihood of hallucinations. This trade-off is evident in our main results (Table 1), where CD-based methods applied to discrete tokenizer-based LVLMs consistently underperform on the CHAIR metric—indicating more hallucinations—despite gains in coverage. We further illustrate this issue through case studies in Figure 8. For instance, SID [2] and VCD often extend responses unnecessarily, even when the initial answer is sufficient. In the first example, SID redundantly restates the scene ("The image displays...") after the model has already completed a coherent description.

## D.5 Evaluation on POPE

POPE (Polling-based Object Probing Evaluation) [19] is a benchmark designed to systematically evaluate object hallucination in LVLMs by formulating the task as a binary classification problem with yes/no questions about objects in images. We perform POPE evaluation on both MSCOCO [26] and GQA [40] datasets under three test settings: Random, Popular, and Adversarial. The evaluation metrics are accuracy and F1 score, which have been presented in Appendix A.1.1.

The results in Table 17 demonstrate that our proposed CGC+VTD method achieves the best performance on 31 out of 36 metrics across different models and datasets and OPERA emerges as the second-best method in most scenarios Notably, the strong performance of CGC+VTD on the GQA dataset demonstrates excellent generalization capabilities, considering that CGC was trained exclusively on the COCO images (including MSCOCO). This cross-dataset transferability suggests that our method captures fundamental aspects of visual priors rather than merely exploiting dataset-specific patterns.

Table 17: Evaluation results on POPE. Best/second best results are marked **bold**/underlined.

| Dataset | Model | Method | Random | | Popular | | Adversarial | |
|---|---|---|---|---|---|---|---|---|
| | | | Accuracy | F1 Score | Accuracy | F1 Score | Accuracy | F1 Score |
| MSCOCO | Chameleon-7B | Nucleus Sampling | 50.6 | 60.0 | 55.00 | 62.4 | 52.7 | 61.2 |
| | | VCD | 51.6 | 62.2 | 51.7 | 62.3 | 50.9 | 61.4 |
| | | SID | 51.4 | 61.0 | 50.9 | 60.7 | 52.2 | 62.1 |
| | | OPERA | 52.6 | **65.8** | 54.2 | 66.1 | 50.2 | 65.7 |
| | | Cluster-Base | 50.6 | 60.9 | 54.9 | 62.2 | 52.5 | 61.1 |
| | | CGC+VTD | **52.7** | 63.9 | **56.5** | **68.0** | **53.0** | **66.9** |
| | Janus-Pro-7B | Nucleus Sampling | 89.3 | 88.4 | 87.5 | 86.7 | 85.2 | 84.7 |
| | | VCD | 71.3 | 63.9 | 70.9 | 63.5 | 68.6 | 61.1 |
| | | SID | 68.9 | 59.3 | 68.0 | 57.7 | 66.3 | 55.8 |
| | | OPERA | 89.7 | 88.9 | 87.8 | **87.2** | 85.4 | 85.0 |
| | | Cluster-Base | 88.2 | 87.0 | 86.6 | 85.5 | 84.8 | 83.9 |
| | | CGC+VTD | **89.8** | **89.2** | **88.2** | 86.9 | **85.9** | **86.2** |
| | Emu3-13B | Nucleus Sampling | 87.4 | 86.0 | 85.5 | 84.1 | 83.9 | 82.7 |
| | | VCD | 87.0 | 85.6 | 84.5 | 83.2 | 83.3 | 82.2 |
| | | SID | 87.5 | 86.1 | 85.0 | 83.7 | 83.0 | 81.9 |
| | | OPERA | 87.9 | 85.3 | 86.4 | 85.4 | 84.2 | **83.8** |
| | | Cluster-Base | 87.3 | 85.8 | 85.4 | 83.9 | 83.8 | 82.5 |
| | | CGC+VTD | **88.3** | **87.0** | **87.1** | **86.1** | **84.9** | **83.8** |
| GQA | Chameleon-7B | Nucleus Sampling | 53.2 | 62.5 | 57.6 | 64.8 | 55.4 | 63.9 |
| | | VCD | 54.3 | 64.6 | 54.5 | 64.7 | 53.6 | 63.8 |
| | | SID | 54.1 | 63.5 | 53.7 | 63.2 | 54.9 | 64.5 |
| | | OPERA | 55.2 | **68.2** | 56.9 | **68.6** | 52.9 | 68.1 |
| | | Cluster-Base | 53.3 | 63.4 | 57.4 | 64.6 | 55.1 | 63.7 |
| | | CGC+VTD | **55.4** | 66.4 | **59.1** | **70.4** | **55.7** | **69.3** |
| | Janus-Pro-7B | Nucleus Sampling | 91.7 | 90.9 | 89.9 | 89.2 | 87.8 | 87.3 |
| | | VCD | 74.0 | 66.7 | 73.6 | 66.2 | 71.3 | 63.9 |
| | | SID | 71.6 | 62.1 | 70.7 | 60.6 | 69.0 | 58.7 |
| | | OPERA | 92.2 | 91.4 | 90.3 | **89.7** | 87.9 | 87.5 |
| | | Cluster-Base | 90.8 | 89.7 | 89.2 | 88.1 | 87.4 | 86.5 |
| | | CGC+VTD | **92.3** | **91.7** | **90.7** | 89.4 | **88.5** | **88.8** |
| | Emu3-13B | Nucleus Sampling | 90.0 | 88.7 | 88.1 | 86.9 | 86.5 | 85.4 |
| | | VCD | 89.6 | 88.3 | 87.1 | 86.0 | 85.9 | 84.9 |
| | | SID | 90.1 | 88.8 | 87.6 | 86.5 | 85.6 | 84.6 |
| | | OPERA | 90.5 | 88.0 | 89.0 | 88.1 | **87.8** | 86.4 |
| | | Cluster-Base | 89.9 | 88.5 | 88.0 | 86.7 | 86.4 | 85.2 |
| | | CGC+VTD | **90.9** | **89.7** | **89.7** | **88.8** | 87.5 | **86.5** |

