# OpenReview forum: "Image Token Matters: Mitigating Hallucination in Discrete Tokenizer-based Large Vision-Language Models via Latent Editing"
_NeurIPS.cc/2025/Conference — NeurIPS 2025 poster_

### Official Review · Reviewer_WjUQ · 2025-06-22

**Clarity:** 3
**Significance:** 3
**Originality:** 3
**Rating:** 5
**Confidence:** 4

**Summary:**

The paper proposes a novel hallucination mitigation method specifically designed for LVLMs based on discrete image tokenizers. The method is based on first constructing a co-occurrence graph to characterize image tokens that frequently co-occur based on their spatial proximity and semantic objects they are associated to. It then uses a GNN to extract image token embeddings and clusters them to represent the co-occurring tokens. Hallucinated objects are associated to tokens of the most dominant ("commonly present") clusters that are not detected in the current image. Finally, during inference, latent representations are edited to reduce influence of of all such tokens.

**Questions:**

1. How did you arrive to the 3x3 grid division for determining the spatial proximity. I expect it might not make a major difference but did you experiment with other choices?

2. I am not entirely convinced by the need to use GNNs for extracting token representations. Did you try using a simple way of extracting node representations for clustering, for instance, just a vector summarising the edge strengths of a node? I am also not sure if line 237 references the same baseline. What representation did you use there for clustering?

3. Line 315: Could you please describe more precisely the combination of CGC with other methods.

4. If you performed a simple clustering step on continuous visual embeddings to manually discretize, does it not allow the method to also easily generalise to LVMLs using continuous visual tokens.

**Ethical Concerns:**

["NO or VERY MINOR ethics concerns only"]

**Final Justification:**

1. I already found the paper solid with no major weakness and was inclined to score it 5
2. The authors adequately responded to my questions
3. In my opinion, the only major concern I have seen in other reviews is that the method does not extend to LVLMs using continuous image tokens. Thus it has limited value. While an understandable concern, the authors have been very upfront about it in the paper from the beginning. To me the contribution is still valuable.

Overall, I will increase my score to 5. At the moment, I am still open to discussion with other reviewers/AC. So this score could still change until deadline (Aug 13).

**Limitations:**

I am not sure the method generalizes to architectures that use Q-Former to process vision tokens even if discrete image tokens, or LVLMs which integrate vision modality through cross attention maps at all layers.

**Quality:**

3

**Strengths And Weaknesses:**

Strengths:

- The writing and paper presentation is very clear in most parts.
- The problem is relevant, contemporary and the proposed method is also clearly novel.
- Experimental evaluation is very solid, showing consistent improvement on multiple datasets, models and against multiple relevant baselines.

Weaknesses:

I do not believe the paper has any major weakness. I would like to rate the paper as 5, However, I do have doubts regarding certain design choices or experiments in the paper. Please refer to the questions section for the same.

---

> ### Author Rebuttal · Authors · 2025-07-31
>
> We appreciate your positive feedback about **high novelty, clear presentation, and solid experiments**. We address your questions below.
>
> ---
>
> Q1: How did you arrive to the 3x3 grid division for determining the spatial proximity. I expect it might not make a major difference but did you experiment with other choices?
>
> A1: We have indeed explored other choices. We evaluated grid sizes from 1×1 to 6×6 on Janus-Pro-7B using the Object HalBench benchmark:
> | **Metric** | **1x1** | **2x2** | **3x3** | **4x4** | **5x5** | **6x6** |
> |------------|---------|---------|---------|---------|---------|---------|
> | CHAIR-s    | 14.95   | 10.78   | 10.43   | 12.52   | 12.11   | 13.27   |
> | CHAIR-i    | 9.94    | 6.11    | 5.81    | 6.19    | 6.32    | 7.01    |
>
> The 3×3 grid achieves optimal performance, though 2×2 to 6×6 all provide reasonable results within acceptable ranges. The 1×1 setting performs poorly due to insufficient edge connectivity during graph construction, which impairs GNN learning quality by creating overly sparse co-occurrence relationships.
>
> ---
>
> Q2: I am not entirely convinced by the need to use GNNs for extracting token representations. Did you try using a simple way of extracting node representations for clustering, for instance, just a vector summarising the edge strengths of a node? I am also not sure if line 237 references the same baseline. What representation did you use there for clustering?
>
> A2: The GNN learning step is essential, as it is widely recognized that GNNs are effective for node clustering due to their ability to learn node embeddings that capture graph structural information. In our case, the co-occurrence graph is constructed based on spatial proximity and semantic coherence, so effectively modeling its structure with a GNN amounts to capturing visual priors. We have shown in Table 1 that trivial clustering methods like naive K-means on the original codebook representations (Cluster-Base method) yield substantially inferior performance compared to our CGC+VTD approach. This supports our decision to employ GNNs.
>
> The "baseline" method referenced in line 237 uses naive K-means clustering directly on the original VQVAE codebook embeddings without any graph-based representation learning.
>
> ---
>
> Q3: Line 315: Could you please describe more precisely the combination of CGC with other methods.
>
> A3: We have provided details of how to combine CGC with other methods in Appendix B.2. Due to the page limit, we could not put them in the main body of our submission. We hope this can address your concern.
>
> ---
>
> Q4: If you performed a simple clustering step on continuous visual embeddings to manually discretize, does it not allow the method to also easily generalise to LVMLs using continuous visual tokens.
>
> A4: This is a great question that we have also considered. While the intuition appears straightforward, the practical implementation presents significant technical challenges. Our method fundamentally relies on identifying "absent tokens" from a finite, discrete token vocabulary. Adopting continuous visual embeddings can be understood as introducing infinite visual tokens, making it impossible to get the set of "absent tokens" anymore.
>
> ---
>
> Limitation: I am not sure the method generalizes to architectures that use Q-Former to process vision tokens even if discrete image tokens, or LVLMs which integrate vision modality through cross attention maps at all layers.
>
> A5: While architectures like Q-Former and intra-layer cross attention were important early milestones in LVLM development, they fall outside the scope of our work. Our work focuses on a new class of LVLMs that adopt a unified token processing paradigm, where discrete image tokens are processed directly alongside text tokens within a single transformer architecture. In these models, there is typically no need for modality-specific components such as Q-Formers (as in BLIP variants), intra-layer cross-attention(as in Flamingo), or lightweight MLP adapters (as in LLaVA), since discrete tokenization naturally supports seamless cross-modal interaction without explicit alignment mechanisms. As the design space for unified LVLMs continues to evolve, our work offers foundational insights into hallucination behavior grounded in visual priors. These insights may help inform intervention strategies across future architectures, and we would be happy to engage in further discussion if you are interested.

---

> > ### Comment · Reviewer_WjUQ · 2025-08-02
> > **Rebuttal acknowledgement**
> >
> > Thank you to the authors for the rebuttal. It addresses my questions adequately.

---

> > > ### Author Response · Authors · 2025-08-05
> > >
> > > We sincerely appreciate your engagement in the discussion and your positive assessment of our paper.
> > > We will incorporate the insights from our discussions to further strengthen and improve the paper.

---

### Official Review · Reviewer_qYkC · 2025-06-23

**Clarity:** 3
**Significance:** 3
**Originality:** 4
**Rating:** 4
**Confidence:** 3

**Summary:**

This paper introduces a framework for modeling visual priors in LVLMs equipped with discrete image tokenizers. Motivated by the observation that certain image tokens frequently co-occur to represent common objects or scenes, the authors construct a co-occurrence graph of codebook entries and apply a GNN-based clustering method to uncover dominant token groups. By clustering over codebook embeddings rather than raw token embeddings, the method ensures both computational efficiency and improved semantic coherence. The resulting clusters are shown to be closely associated with hallucination-prone tokens, and the authors further propose a targeted mitigation strategy that edits intermediate representations based on dominant clusters.

**Questions:**

See Weakness section. I would consider raising the score if the identified concerns are properly addressed.

**Ethical Concerns:**

["NO or VERY MINOR ethics concerns only"]

**Final Justification:**

The authors’ response greatly clarified my understanding. Since all of my concerns have been addressed, I will keep my score unchanged.

**Limitations:**

yes

**Paper Formatting Concerns:**

There are no formatting issues.

**Quality:**

3

**Strengths And Weaknesses:**

Strengths:

[1] The paper clearly motivates the concept of visual priors and effectively models them through GNN-based clustering over image tokens derived from a discrete tokenizer.

[2] It also provides a compelling rationale for clustering over codebook embeddings rather than raw image token embeddings, highlighting both efficiency and semantic coherence.

Weaknesses:

[1] Regarding Section 3.2, please consider providing examples of the Top-K(C) object categories for each group—such as C1, C2, and C3—or examples of the tokens contained in each group, along with corresponding hallucinated objects (obj′) and segmentation annotations from the COCO panoptic dataset. The current experimental setup is somewhat difficult to follow without concrete illustrations.

[2] In Section 3.3, please consider using a term like v_candidate instead of v_hal, as the current terminology may be somewhat misleading. This is especially relevant since hallucinated tokens cannot always be explicitly identified in certain benchmarks (e.g., MME or Object HalBench), unlike in AMBER.

---

> ### Author Rebuttal · Authors · 2025-07-31
>
> Thank you for the positive evaluation of the **motivation and our proposed method**. We will address your questions below.
>
> ---
>
> W1: Regarding Section 3.2, please consider providing examples of the Top-K(C) object categories for each group—such as C1, C2, and C3—or examples of the tokens contained in each group, along with corresponding hallucinated objects (obj′) and segmentation annotations from the COCO panoptic dataset. The current experimental setup is somewhat difficult to follow without concrete illustrations.
>
> A1: Consider the following example. For the third case in Figure 8, the top-5 object sets associated with $C_1$,$C_2$ and $C_3$ are: $\text{Top-5} (C_1)=('table', 'sand', 'towel','pizza','cake')$, $\text{Top-5} (C_2)=('sand','bowl','table','cake','water')$, $\text{Top-5} (C_3)=('skateboard','table','ski','knife','light')$. In here, the hallucinated object 'bowl' is the second most significant object associated with $C_2$ and does not appear in the top-5 objects associated with $C_1$ and $C_3$. This demonstrates our central finding: absent tokens from dominant clusters ($C_2$) show stronger correlations with hallucinated content (the object "bowl" in this case) than present tokens or tokens from non-dominant clusters. Please note that although $\text{Top-5}(C_2)$ also contains several ground truth objects, removing $C_2$ tokens' influence primarily eliminates hallucination-prone associations while preserving essential visual information through $C_1$ tokens. This selective intervention maintains recall for actual visual content while reducing false positive generations. We will include this example in our revision for better readability.
>
> To supplement, we also present here how to derive the order of Top-K(C). We first construct a token-object association dictionary for each image token using the COCO panoptic dataset, where we compute the frequency of this token with which it is used to represent objects from different segmentation classes. A token is considered to represent an object if it lies within the segmented area corresponding to that object. The association dictionaries of all image tokens are then aggregated within token groups ($C_1$, $C_2$, $C_3$) to summarize how frequently tokens in each group are used to represent different segmentation classes, based on which we construct Top-K (C) sets ranked by association frequency.
>
> ---
>
> W2: In Section 3.3, please consider using a term like v_candidate instead of v_hal, as the current terminology may be somewhat misleading. This is especially relevant since hallucinated tokens cannot always be explicitly identified in certain benchmarks (e.g., MME or Object HalBench), unlike in AMBER.
>
> A2: Thank you for your suggestion! We will update our notations in the revision.

---

### Official Review · Reviewer_ecat · 2025-07-01

**Clarity:** 4
**Significance:** 3
**Originality:** 3
**Rating:** 4
**Confidence:** 3

**Summary:**

This paper addresses the hallucination issue in large vision-language models (LVLMs) with discrete image tokenizers, proposing a two-stage framework: Context-Guided Clustering (CGC) and Visual Token Decontamination (VTD). CGC captures visual priors by constructing a co-occurrence graph of image tokens and leveraging graph neural networks, while VTD mitigates hallucinations via latent space editing during generation. Experiments on benchmarks like AMBER and Object HalBench demonstrate the method’s effectiveness, but the comparison settings and baseline models have notable limitations.

**Questions:**

1. Baseline relevance: Please verify how the proposed method compares to the SOTA model (e.g. Qwen2.5-VL) for comprehension tasks.
2. Benchmark scope: Are existing benchmarks sufficient to evaluate the baseline model's hallucination in generation tasks? Please verify on a generation-centric dataset.

**Ethical Concerns:**

["NO or VERY MINOR ethics concerns only"]

**Final Justification:**

The authors address my concerns on discrete tokenizer-based LVLMs.

**Limitations:**

yes

**Quality:**

3

**Strengths And Weaknesses:**

Strengths:
1. Novel problem framing: Identifies visual priors from token co-occurrence patterns as a key source of hallucination in discrete-token LVLMs, filling a research gap.
2. Integrated methodology: CGC combines spatial and semantic contexts to model token relationships, and VTD provides a targeted latent editing strategy, forming a cohesive solution.
3. Comprehensive evaluation: Experiments compare multiple baselines across multiple benchmarks, showing consistent hallucination reduction without sacrificing model expressiveness.

Weaknesses:
1. Inadequate baseline fairness: Compared methods (e.g., VCD, SID, OPERA) are primarily designed for understanding-focused LVLMs, not tailored to discrete tokenizer architectures. This may lead to biased comparisons, as contrastive decoding methods, for example, may conflict with discrete token representations.
2. Suboptimal baseline models: Models that unify understanding and generation, such as Chameleon, Emu3, and Janus-Pro, use discrete tokenizers and perform worse than current SOTA LVLMs (e.g., Qwen2.5-VL, Intern3-VL) in visual language understanding tasks. The benchmarks only focus on understanding tasks and lack evaluation on generation tasks.

---

> ### Author Rebuttal · Authors · 2025-07-31
>
> Thanks for your valuable feedback. We appreciate that the reviewer found that our paper proposed a **novel problem and a cohesive methodology**. We address your concerns below.
>
> ---
> W1: Inadequate baseline fairness: Compared methods (e.g., VCD, SID, OPERA) are primarily designed for understanding-focused LVLMs, not tailored to discrete tokenizer architectures. This may lead to biased comparisons, as contrastive decoding methods, for example, may conflict with discrete token representations.
>
> A1: To the best of our knowledge, we are the first to investigate hallucination detection and mitigation in LVLMs using discrete image tokenizers. Existing state-of-the-art methods were all developed for LVLMs with continuous image encoders, and our experiments have shown them to be ineffective when applied to models with discrete tokenizers. We believe our experimental setup is fair. As you noted, the failure of methods such as contrastive decoding may stem from a fundamental mismatch with discrete token representations. **This is not a result of unfair evaluation, but rather a limitation inherent to methods designed for continuous encoders. Our findings highlight this gap and constitute a meaningful contribution, rather than a flaw**. We appreciate your feedback and welcome further discussion.
>
> ---
>
> W2: Suboptimal baseline models: Models that unify understanding and generation, such as Chameleon, Emu3, and Janus-Pro, use discrete tokenizers and perform worse than current SOTA LVLMs (e.g., Qwen2.5-VL, Intern3-VL) in visual language understanding tasks. The benchmarks only focus on understanding tasks and lack evaluation on generation tasks.
>
> A2: LVLMs with discrete tokenizers have recently attracted rapidly growing attention, making it important to investigate their behavior with respect to hallucination. In this paper, we focus exclusively on LVLMs with discrete image tokenizers. Models such as Qwen2.5-VL and Intern3-VL rely on continuous image encoders and are therefore beyond the scope of our study.
>
> **Hallucination in VLMs, as defined by previous research [1, 2, 3, 4], is characterized by VLMs generating textual descriptions that include objects not present in or inconsistent with the target images**. The benchmarks we employ (AMBER, Object HalBench) are established and well-adopted standards in the hallucination-related literature. We strictly follow previous works and use these benchmarks to ensure fair comparison.
>
> We are not certain what you mean by “generation tasks”. If you are referring to image generation tasks, they fall outside the scope of our paper, as they conflict with how multimodal hallucination has been defined in prior work, focusing on visual understanding. If you are referring to text generation, multimodal hallucinations are identified based on the generated textual output, as defined in prior works. Our setting follows this established definition. We are happy to address any additional concerns you may have.
>
>     [1]: Rohrbach, Anna, et al. "Object hallucination in image captioning." arXiv preprint arXiv:1809.02156 (2018).
>     [2]: Li, Yifan, et al. "Evaluating Object Hallucination in Large Vision-Language Models." Proceedings of the 2023 Conference on Empirical Methods in Natural Language Processing. (2023)
>     [3]: Wang, Junyang, et al. "Amber: An llm-free multi-dimensional benchmark for mllms hallucination evaluation." arXiv preprint arXiv:2311.07397 (2023).
>     [4]: Liu, Jiazhen, et al. "PhD: A ChatGPT-Prompted Visual Hallucination Evaluation Dataset" Proceedings of the IEEE/CVF Conference on Computer Vision and Pattern Recognition (CVPR), (2025).
>
> ---
>
> Q1: Baseline relevance: Please verify how the proposed method compares to the SOTA model (e.g. Qwen2.5-VL) for comprehension tasks.
>
> A3: **Our experimental design follows standard practices in hallucination mitigation research, where the appropriate comparison is between different mitigation methods applied to the same base model, not across different architectures.** Comparing against Qwen2.5-VL would be out of the scope of this paper. As clearly stated in our abstract and introduction, we specifically target LVLMs with discrete image tokenizers and do not cover models with continuous image encoders, such as Qwen2.5-VL.
>
> ---
>
> Q2: Benchmark scope: Are existing benchmarks sufficient to evaluate the baseline model's hallucination in generation tasks? Please verify on a generation-centric dataset.
>
> A4: As we clarified in the previous question, hallucination of VLM happens where models generate plausible but incorrect textual descriptions of visual content. The benchmarks we employ (AMBER, Object HalBench) are specifically designed and widely adopted for evaluating hallucinations in visual understanding tasks. Visual generation hallucinations (where models produce incorrect visual content) represent a fundamentally different research problem, which is beyond our research scope.

---

### Official Review · Reviewer_eq2o · 2025-07-02

**Clarity:** 2
**Significance:** 2
**Originality:** 3
**Rating:** 3
**Confidence:** 3

**Summary:**

This paper proposes a novel method, named CGC, to analyze the impact of relationships - including co-occurrence and semantic coherence - between discrete visual words on the hallucination ratio of LVLM models. Based on the analysis results, the authors further propose a solution, named VTD, which reduces the hallucination ratio during decoding by subtracting feature components that may cause hallucinations. Extensive experiments are conducted to demonstrate the effectiveness of the proposed methods.

**Questions:**

1. In Eq. 2 and Eq. 3, I am curious about the intuition behind why the contrastive loss alone is not strong enough, necessitating the introduction of an additional loss to constrain positive similarities. Could the authors elaborate on this point?

2. In Figure 3, group C1 refers to tokens from the dominant cluster(s) that appear in the image. How do these tokens cause hallucinations if they are indeed present in the image?

3. Section 3.2 should be described more clearly. How does the COCO segmentation mask match the AMBER annotations? Can we ensure that all objects contained in AMBER images fall within the predefined COCO segmentation classes?

4. In Eq. 4, since $\hat{g}^{(l)} (v_{hal})$ has been normalized using L2-Norm according to Line 255, can I understand that the denominator in Eq. 4 will always be 1?

**Ethical Concerns:**

["NO or VERY MINOR ethics concerns only"]

**Final Justification:**

I still find the proposed framework overly complex, and I believe its methodological significance remains limited. Therefore, I will maintain my original score.

**Limitations:**

Yes.

**Paper Formatting Concerns:**

No.

**Quality:**

3

**Strengths And Weaknesses:**

### Strengths
1. There is currently a lack of sufficient discussion about the causes of hallucinations in VLMs, and this paper makes a valuable contribution to addressing this gap.
2. The authors conduct extensive experiments to demonstrate the effectiveness of their proposed methods.

### Weaknesses
1. The significance of the proposed method is a bit limited, as most VLMs now use continuous visual tokens to achieve better perception and cognition performance. As mentioned in the limitations section, the method can only be applied to VLMs using discrete tokens, which are mostly designed for unified understanding and generation models.
2. Some points could be further clarified; please refer to the “Questions” section for details.
3. The overall writing, organization, and illustrations could be improved to enhance clarity and readability.

---

> ### Author Rebuttal · Authors · 2025-07-30
>
> We appreciate your positive feedback on our **motivation and solid experiments**. We hope our answers can address your concerns and questions.
>
> ---
>
> W1: The significance of the proposed method is a bit limited...
>
> A1: Recent advances in LVLMs have increasingly focused on unifying understanding and generation within single frameworks [1,2,3], with discrete image tokenizers emerging as a particularly promising approach [4,5,6]. The recent success of BAGEL [4], which outperforms Qwen2.5-VL on 6 out of 7 image understanding benchmarks, demonstrates the potential of this unified architecture paradigm.
>
> We acknowledge that most previous hallucination mitigation research on LVLMs is based on continuous LVLMs; however, as LVLMs with discrete image tokenizers are gaining more popularity and have shown extraordinary performance, we believe it is very important to systematically study hallucination mitigation specifically for discrete tokenizer-based LVLMs. To our knowledge, we are the first to identify and quantify hallucination sources arising from discrete image token co-occurrence patterns, providing a novel perspective for understanding visual hallucinations in unified multimodal architectures. Our proposed CGC+VTD framework directly addresses these visual priors through targeted latent space intervention, achieving superior performance while maintaining computational efficiency.
>
>     [1] Zhang, Xinjie, et al. "Unified multimodal understanding and generation models: Advances, challenges, and opportunities." arXiv preprint arXiv:2505.02567 (2025).
>     [2] Li, Zijie, et al. "Dual diffusion for unified image generation and understanding." Proceedings of the Computer Vision and Pattern Recognition Conference. 2025.
>     [3] Qu, Liao, et al. "Tokenflow: Unified image tokenizer for multimodal understanding and generation." Proceedings of the Computer Vision and Pattern Recognition Conference. 2025.
>     [4] Deng, Chaorui, et al. "Emerging properties in unified multimodal pretraining." arXiv preprint arXiv:2505.14683 (2025).
>     [5] Lin, Haokun, et al. "Toklip: Marry visual tokens to clip for multimodal comprehension and generation." arXiv preprint arXiv:2505.05422 (2025).
>     [6] Xie, Jinheng, et al. "Show-o2: Improved Native Unified Multimodal Models." arXiv preprint arXiv:2506.15564 (2025).
>
> ---
>
> W3: The overall writing, organization, and illustrations could be improved to enhance clarity and readability.
>
> A2: Thanks for the suggestion. We will improve the clarity and readability in the revision.
>
> ---
>
> Q1: In Eq. 2 and Eq. 3, I am curious about the intuition behind why the contrastive loss alone is not strong enough...
>
> A3: Our core intuition is that while  $\mathcal{L}\_{\text{contrast}}$ uses co-occurrence weights for relative optimization, **$\mathcal{L}\_{\text{pps}}$ enforces absolute similarity thresholds that ensure adequate separation between positive and negative pairs** regardless of batch composition. Concretely, the gradient of $\mathcal{L}\_{\text{contrast}}$ w.r.t. $\text{sim}(\mathbf{h}\_i, \mathbf{h}\_j)$ is positively correlated to $s\_{ij}$ while negatively correlated to the number of negative sample pairs. So the pressure to improve the similarity between node i and node j towards a high $s\_{ij}$ could be diminished by a large number of negative pairs within the batch. $\mathcal{L}\_{\text{pps}}$ addresses this by setting explicit minimum similarity targets ($\beta \cdot s_{ij}$) that must be satisfied independently of other pairs in the batch. To further address your concern, we conducted additional analysis on the similarity patterns during GNN training using Janus-Pro-7B on the COCO 2017 validation set.
>
> | **Variant** | **avg. positive similarity** | **avg. negative similarity** | **avg. similarity gap** |
> |-------------|------------------------------|------------------------------|--------------------------|
> | w/o. $\mathcal{L}_\text{pps}$ | 0.56 | 0.35 | 0.21 |
> | w. $\mathcal{L}_\text{pps}$ | 0.79 | 0.46 | 0.33 |
>
> Without $\mathcal{L}\_{\text{pps}}$, the model achieves only 0.56 average positive similarity with a similarity gap of 0.21 between positive and negative pairs. With $\mathcal{L}\_{\text{pps}}$, positive similarity increases substantially to 0.79, while the similarity gap expands to 0.33 by 57%. This demonstrates that $\mathcal{L}\_{\text{pps}}$ successfully enforces the absolute similarity constraints, ensuring that co-occurrence relationships are preserved with sufficient fidelity for downstream clustering and hallucination mitigation. Furthermore, the ablation study in Table 4 consistently shows performance degradation when $\mathcal{L}\_{\text{pps}}$ is removed, confirming that $\mathcal{L}\_{\text{pps}}$ is crucial for our method.
>
> ---
> Q2: In Figure 3, group C1 refers to tokens from the dominant cluster(s) that appear in the image. How do these tokens cause hallucinations if they are indeed present in the image?
>
> A4: **C1 tokens do not directly cause hallucinations, but they can be associated with hallucinated objects due to the limited vocabulary size of discrete tokenizers**. Since discrete tokenizers use a fixed vocabulary to represent diverse visual content, each token must encode multiple visual concepts and objects. Consequently, any given token—including those present in the image (C1)—will be associated with various objects, encompassing both those that genuinely appear in the current image and those that do not. Our analysis demonstrates that while C1 tokens may indeed correlate with some hallucinated objects, the absent C2 tokens from the same dominant cluster show significantly stronger correlations with hallucinated content. This finding reveals that hallucinations primarily arise from the model's inappropriate activation of absent tokens that frequently co-occurred with present ones during training, rather than from the present tokens themselves. Therefore, C1 tokens are not the direct cause of hallucinations—they legitimately represent visual content in the image. The hallucinations stem from the learned co-occurrence priors that cause the model to inappropriately reference their absent cluster partners (C2 tokens). This is why our Visual Token Decontamination method specifically targets C2 tokens for suppression while preserving the contributions of present C1 tokens.
>
> ---
> Q3: Section 3.2 should be described more clearly....
>
> A5: For the analysis in Section 3.2, we only consider the AMBER objects that are covered by COCO segmentation classes. We do this analysis as follows. We first construct a token-object association dictionary for each image token using the COCO panoptic dataset, where we compute the frequency of this token with which it is used to represent objects from different segmentation classes. A token is considered to represent an object if it lies within the segmented area corresponding to that object. The association dictionaries of all image tokens are then aggregated within token groups (C1, C2, C3) to summarize how frequently tokens in each group are used to represent different segmentation classes, based on which we construct Top-K (C) sets ranked by association frequency. To compute HitRate@K, we extract hallucinated objects identified by the AMBER and use string matching against COCO segmentation classes to determine whether each hallucinated object appears within the Top-K(C) set.
>
> Although this approach may overlook the hallucinations linked to AMBER objects not covered by COCO classes, it still ensures methodological consistency and does not compromise our core findings. The dramatic HitRate difference between C2 and other groups remains statistically significant, validating our hypothesis about visual priors.
> To further address your concern, we conduct another HitRate analysis using 500 images from the COCO dataset with Janus-Pro-7B. We can observe that C2 has a much higher HitRate compared with using AMBER, and the huge gap between C2 and other groups (C1 and C3) remains.
>
> | **Token group** | HitRate@2 | HitRate@4 | HitRate@6 | HitRate@8 | HitRate@10 |
> |-----------------|-----------|-----------|-----------|-----------|------------|
> | C2              | 0.18      | 0.26      | 0.33      | 0.33      | 0.37       |
> | C1              | 0.08      | 0.11      | 0.11      | 0.14      | 0.16       |
> | C3              | 0.06      | 0.09      | 0.11      | 0.16      | 0.20       |
>
> Moreover, it i is important to emphasize that visual priors extend beyond object-level associations—they also capture spatial relationships where tokens frequently appear in proximity within images. Our CGC method deliberately incorporates both spatial co-occurrence (tokens appearing within the same local regions) and semantic co-occurrence (tokens representing the same segmented objects) to comprehensively model these visual priors at the token level.
>
> The HitRate analysis serves as an illustrative proxy to demonstrate the effectiveness of our clustering approach. Our method fundamentally operates on token-level co-occurrence patterns rather than being limited to object-level associations. This token-centric approach enables our method to capture richer visual relationships that go beyond simple object categories.
> The strong performance of our method across different datasets also validates its generalizability. As shown in Table 1, our approach trained on COCO data effectively mitigates hallucinations on AMBER, which has a different data distribution. This cross-dataset effectiveness demonstrates that our method captures fundamental visual co-occurrence patterns rather than dataset-specific artifacts, making it broadly applicable to diverse evaluation scenarios.
>
> ---
>
> Q4: In Eq. 4, since $g^{(l)}(v_\text{hal})$ has been normalized using L2-Norm according to Line 255, can I understand that the denominator in Eq. 4 will always be 1?
>
> A6: Yes, your understanding is correct. We will simplify this equation in the revision.

---

> > ### Comment · Reviewer_eq2o · 2025-08-04
> > **Reply to authors' response**
> >
> > Thank you to the authors for the detailed response to my concerns and questions. I agree with the authors’ observation that recent LVLM research is increasingly focused on unifying vision understanding and generation within a single framework. However, I would like to point out that the cited works (BAGEL and Dual-Diffusion) are both designed for continuous vision features, which limits their relevance as supporting examples for the proposed approach.
> >
> > Overall, I still find the proposed framework overly complex, and I believe its methodological significance remains limited. Therefore, I will maintain my original score.

---

> > > ### Author Response · Authors · 2025-08-04
> > >
> > > We appreciate your continued engagement and the confirmation of the rising trend of unified VLMs.
> > >
> > > First, we sincerely apologize for the typo in the citations during our previous rebuttal. What we originally intended to convey is that BAGEL and Dual-Diffusion are used to support the advancing development of unified VLMs in general. Unified VLMs with discrete image tokenizers like MUSE-VL [1] also perform on par with Qwen2.5 VL in various visual understanding benchmarks. Liquid [2], Tokenflow [3], and Unitok [4] also demonstrate promising performance. We believe investigating discrete tokenization merits attention, given its growing adoption in recent models. We appreciate your thoughtful consideration and hope this response effectively addresses your concern.
> > >
> > > [1]: Xie, Rongchang, et al. "Muse-vl: Modeling unified vlm through semantic discrete encoding." arXiv preprint arXiv:2411.17762 (2024).
> > >
> > > [2]: Wu, Junfeng, et al. "Liquid: Language models are scalable and unified multi-modal generators." arXiv preprint arXiv:2412.04332 (2024).
> > >
> > > [3]: Qu, Liao, et al. "Tokenflow: Unified image tokenizer for multimodal understanding and generation." Proceedings of the Computer Vision and Pattern Recognition Conference. 2025.
> > >
> > > [4]: Ma, Chuofan, et al. "Unitok: A unified tokenizer for visual generation and understanding." arXiv preprint arXiv:2502.20321 (2025).
> > >
> > > ---
> > >
> > > On methodological significance: Our framework is specifically designed to capture and remove visual priors in discrete image tokens, which is **a previously unexplored hallucination source**. Our work follows a rigorous three-step logical progression:
> > > 1. We assume that hallucinations stem from visual priors encoded in token co-occurrence patterns.
> > > 2. We design CGC to detect these patterns and empirically validate our assumption by demonstrating that absent tokens from dominant clusters show much higher correlation with hallucinated objects.
> > > 3. We develop VTD to effectively mitigate hallucinations from this identified source.
> > > This complete scientific framework, including hypothesis, validation, and solution, represents a significant methodological contribution that reveals and addresses a fundamental issue in discrete tokenization.
> > >
> > > ---
> > >
> > > On complexity: Each component of our method serves a distinct and clear purpose: CGC identifies co-occurrence patterns that standard clustering cannot detect, and VTD targets specific tokens that drive hallucinations. **This methodology is not redundant but essential for addressing visual prior-driven hallucinations, representing the minimal complexity needed to solve this novel problem.** Moreover, while CGC requires offline training once per model, VTD introduces little computational overhead during inference and thus is up to 5 times faster than the current methods. Also, our method can be orthogonally combined with existing methods, showing great compatibility.
> > >
> > > We hope our answer can address your concerns, and we welcome further discussion.

---

> > > ### Author Response · Authors · 2025-08-06
> > >
> > > Thank you again for your valuable feedback on our manuscript and your engagement in the rebuttal. Please let us know if we have adequately addressed your concerns in our follow-up responses and whether you have any additional feedback. We value your expertise and welcome continued dialogue.

---

### Comment · Area_Chair_H7rc · 2025-08-06
**DDL is approaching the end**

Dear Reviewers, ﻿

Could you kindly review the authors’ rebuttal as well as the comments from your fellow reviewers, and share your thoughts on the authors’ responses? Many thanks. ﻿

Best regards,

AC

---

### Note · Authors · 2025-08-12

Dear Reviewers, AC, and SAC,

We sincerely thank you for reviewing our paper and for overseeing the review process.

We are grateful for the positive feedback received:

**New problem and novel perspective**: The paper is the first to study hallucination mitigation in discrete image tokenizer-based LVLMs. It provides a new perspective to identify and model visual priors from discrete image token co-occurrence patterns as a source of hallucination. (Reviewers: ecat, eq2o, WjUQ)

**Clear motivation and novel methodology**: The paper clearly motivates the concept of visual priors and effectively models them through GNN-based clustering over codebook embeddings, followed by latent space editing for hallucination mitigation. (Reviewers: qYkC, WjUQ)

**Comprehensive and solid experimental evaluation**: Our method demonstrates consistent improvements across multiple datasets, models, and benchmarks, highlighting both inference-time efficiency and effectiveness. (Reviewers: eq2o, ecat, WjUQ)

During the rebuttal, we believe we have thoroughly addressed reviewers' major concerns:

Reviewer eq2o: We provided a detailed analysis demonstrating the necessity of loss function design and clarified the distinction between present (C1) and absent (C2) tokens.

Reviewer ecat: We clarified our scope focuses specifically on discrete image tokenizer-based LVLMs and that our evaluation follows standards in visual understanding tasks.

Reviewer qYkC: We provided concrete examples of token-object associations.

Reviewer WjUQ: We validated our design choices through ablation studies and explained the necessity of GNN-based representation learning.

Regarding the remaining question from reviewer eq2o that we have already addressed in the discussion phase, we want to again emphasize that our framework represents the minimal complexity needed to solve the problem. Each component serves a distinct purpose: CGC identifies co-occurrence patterns that standard clustering cannot detect, while VTD performs targeted editing. This approach achieves superior performance with high efficiency.

In summary, we believe our paper contributes to the research community by (1) identifying and quantifying a previously unexplored source of hallucination in discrete image tokenizer-based LVLMs, (2) proposing the first systematic solution CGC+VTD that achieves superior performance while maintaining computational efficiency, and (3) conducting extensive experiments to validate our solution.

---

### Decision · Program_Chairs · 2025-09-17

**Decision:**

Accept (poster)

**Comment:**

This paper proposes the CGC method to analyze how co-occurrence and semantic coherence among discrete visual words affect the hallucination rate in LVLMs. Based on the findings, the authors further introduce VTD, which reduces hallucinations during decoding by removing feature components prone to causing them. Extensive experiments demonstrate the effectiveness of the proposed methods.

Pros.

1. This paper identifies a new problem, which studies the hallucination mitigation in LVLMs with discrete image tokenizers.

2. The motivation of the method is clear. The concept of visual priors is effectively modeled by using GNN-based clustering on codebook embeddings.

3. The method of using latent space editing for hallucination reduction is novel.

Cons.

The applicability of the study is limited to some extent. Most existing methodologies used to process the image adopt continuous vision feature, while this paper studies the discrete tokens.

Overall, this paper is novel in its problem and methodology but limited in its applicability.